# Active Offline Policy Selection

**Ksenia Konyushkova**[*]
DeepMind
kksenia@deepmind.com

**Yutian Chen**[*]
DeepMind
yutianc@deepmind.com

**Tom Le Paine**
DeepMind
tpaine@deepmind.com

**Caglar Gulcehre**
DeepMind
caglarg@deepmind.com

**Cosmin Paduraru**
DeepMind
paduraru@deepmind.com

**Daniel J Mankowitz**
DeepMind
dmankowitz@deepmind.com

**Misha Denil**
DeepMind
mdenil@deepmind.com

**Nando de Freitas**
DeepMind
nandodefreitas@deepmind.com

## Abstract

This paper addresses the problem of policy selection in domains with abundant logged data, but with a restricted interaction budget. Solving this problem would enable safe evaluation and deployment of offline reinforcement learning policies in industry, robotics, and recommendation domains among others. Several off-policy evaluation (OPE) techniques have been proposed to assess the value of policies using only logged data. However, there is still a big gap between the evaluation by OPE and the full online evaluation in the real environment. Yet, large amounts of online interactions are often not possible in practice. To overcome this problem, we introduce *active offline policy selection* — a novel sequential decision approach that combines logged data with online interaction to identify the best policy. This approach uses OPE estimates to warm start the online evaluation. Then, in order to utilize the limited environment interactions wisely we decide which policy to evaluate next based on a Bayesian optimization method with a kernel function that represents policy similarity. We use multiple benchmarks with a large number of candidate policies to show that the proposed approach improves upon state-of-the-art OPE estimates and pure online policy evaluation [2].

## 1 Introduction

Reinforcement learning (RL) has recently proven to be successful in a range of applications from computer and board games [46, 55, 58] to robotics and chip design [37, 10, 45]. However, many challenges of real-world systems still prevent RL from being applied at scale in practice [16]. One of the limiting factors in real applications is that environment interactions are often expensive. Thus, training and evaluating agents becomes prohibitively slow and costly. Offline RL attempts to address this problem by training agents on a dataset without environment interactions [2, 23]. This enables researchers to train policies with multiple algorithms and different hyperparameter settings [50] on a single dataset. The best policy can then be chosen with off-policy policy evaluation (OPE) approaches [53, 40] (Fig. 1, left). These techniques attempt to estimate the expected agent performance by relying on the same pre-recorded dataset [28, 41, 36, 72].

---

[*]equal contribution

[2]The paper website is at https://sites.google.com/corp/view/active-ops and the code is at https://github.com/deepmind/active_ops.

35th Conference on Neural Information Processing Systems (NeurIPS 2021).

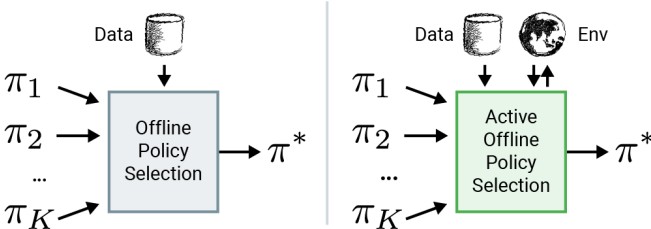

Figure 1: Left: Offline policy selection attempts to choose the best policy from a set of policies, given only a dataset of logged experiences. Right: *Active offline policy selection* additionally assumes a restricted environment interaction budget which is assigned intelligently to evaluate the most promising policies.

Unfortunately, OPE estimates are not yet precise enough to be relied upon for choosing which agent to deploy. Moreover, OPE is intrinsically limited by the pre-recorded dataset: When there is a distribution shift between a trained policy and a behavioural policy, *i.e.*, when the trained policy behaves very differently from the data collection policy, OPE is imprecise [38]. In many applications, however, a subset of policies can be evaluated *online* prior to final deployment [23, 19], even though this process can be hard and laborious. For example, in robotics a small subset of offline RL policies can be tested on the real hardware. Another example is recommender systems, which allow for restricted policy evaluation via A/B tests on a small fraction of user traffic.

In this paper, we introduce the *active offline policy selection* problem. The goal is to identify the best policy where we benefit from both offline evaluation with logged data and online evaluation with limited interaction (Fig. 1, right). In our approach we advance a solution based on Bayesian optimization (BO). It entails learning a Gaussian process (GP) surrogate function that maps policies to their expected returns. Then, we build on the GP statistics to construct an acquisition function to decide which policy to test next. To make BO successful in this problem setting, our approach has two key features. First, we incorporate existing OPE estimates as additional noisy observations (Fig. 2). This allows us to warm start online policy evaluation and to overcome the difficulties of GP hyper-parameter optimisation at the start. Second, we model correlation between policies through a kernel function based on actions that the policies take in the same states of the environment (Fig. 3). This makes our method data efficient [26] as the information about the performance of one policy informs us about the performance of similar behaving policies without costly execution in the environment. It is particularly valuable when the number of candidate policies is large (or even larger than the interaction budget).

The contributions of this paper are as follows.

1. We introduce the *active offline policy selection* problem. This problem is important in practice, but so far it has not been studied in the literature to the best of our knowledge.

2. We build a BO solution with an extended observation model to integrate both OPE estimates and interaction data. The use of OPE estimates enables us to warm-start learning a GP.

3. We propose a novel GP kernel to capture the dependency between the policies through the actions that they take. As a result, our method infers the value of one policy from those of similar policies.

The rest of this paper is organized as follows. Section 1.1 gives background on offline policy evaluation and selection. In section 2, we propose a GP model for policy value estimation, explain how we approach the sequential decision making with BO and introduce a novel kernel based on the similarity of policy outputs. Section 3 discusses the related work. Section 4 shows that active policy evaluation can improve upon the OPE after merely a few interactions, thanks to the kernel that ensures data-efficiency. Additionally, our method works reliably with OPEs of varying quality and it scales well with the growing number of candidate policies. The paper is concluded with a discussion in section 5.

## 1.1 Off-policy policy evaluation and selection

We consider a Markov decision process (MDP) defined by a tuple $(S, A, T, R, d_0, \gamma)$, with state space $S$, action space $A$, transition distribution $T(s'|s, a)$, reward function $R(s, a)$, initial state distribution

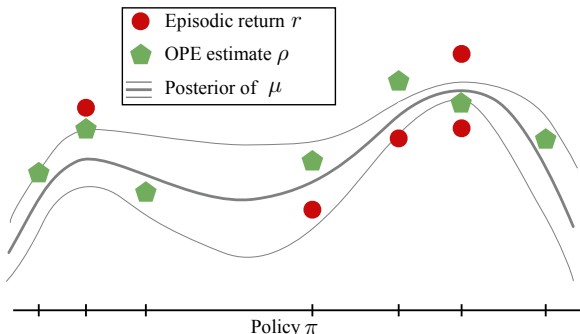

Figure 2: Gaussian process of the policy value function. We observe OPE estimates for all policies and noisy episodic returns for some of them. Neighboring policies (measured by kernel $\mathcal{K}$) have similar values, and the posterior variance is lower where there are more observations. Policies are aligned in 1D for visualization purposes, but in practice they reside in a high dimensional space.

$d_0(s)$, and discount factor $\gamma \in [0, 1]$. A policy $\pi(a|s)$ maps from $S$ to $A$. The value of a policy $\pi$ is measured by the expected sum of discounted rewards:

$$\mu_\pi = \mathbb{E}\left[\sum_{t=0}^{\infty} \gamma^t R(s_t, a_t)\right], \text{ with } s_0 \sim d_0(\cdot), a_t \sim \pi(\cdot|s_t), s_{t+1} \sim T(\cdot|s_t, a_t). \tag{1}$$

Typically, estimating the value of a given policy requires executing the policy in the environment many times. However, practical constraints make it difficult to run many policies of unknown quality in the environment. To address this, OPE estimates a value $\hat{\mu}_\pi$ using a dataset of trajectories $\mathcal{D}$ collected by a separate behavior policy $\pi_\beta$. In offline policy selection (OPS) the task is to select the policy with the highest value from a set of policies $\pi_1, \pi_2, \ldots, \pi_K$ given access only to a dataset $\mathcal{D}$ (Fig. 1, left). A straightforward approach is to select the policy with the highest OPE estimate $\hat{\mu}_{\pi_1}, \ldots, \hat{\mu}_{\pi_K}$ [20], and alternative approaches can be used depending on the quality metric [71].

## 2 Active offline policy selection

### 2.1 Problem definition

We now formally define the active offline policy selection problem. Suppose we are given a set of $K$ candidate policies $\pi_k, 1 \le k \le K$. Denote by $\mu_k$ the unknown policy value and by $\rho_k$ an OPE estimate computed from an offline dataset. At every step $i$, we can choose a policy $k_i$ to execute in the environment once and observe the trajectory with (discounted) episodic return $r_i$, which is a noisy sample of the true policy value: $\mathbb{E}[r_i] = \mu_{k_i}$. We would like to find the policy with the highest value, $k^* = \arg\max_k \mu_k$, with as few trajectories (budget) as possible. Policy selection algorithm estimates the mean as $m_k$ and recommends a policy that maximizes it: $\hat{k} = \arg\max_k m_k$. Simple regret of the recommended policy measures how close we are to the best policy:

$$\text{regret} = \mu_{k^*} - \mu_{\hat{k}}. \tag{2}$$

To solve this sequential decision problem, we introduce a GP over the policy returns and OPE estimates (subsection 2.2), and design the GP kernel over policies to model their correlated returns (subsection 2.3). Then we use these as part of a BO strategy (subsection 2.4).

### 2.2 Gaussian process over policy values

Given a limited interaction budget and a large number of policies, it is impossible to obtain accurate estimates of their values by executing them many times. Fortunately, similar policies tend to have similar values. Thus, propagating the information between them helps to improve the value prediction. Intuitively, the number of policies that we need to execute to find a good policy should depend on their diversity instead of the total number of candidate policies (see experiments supporting this in Fig. 7). Formally, we use GP to predict a joint distribution over the policy values.

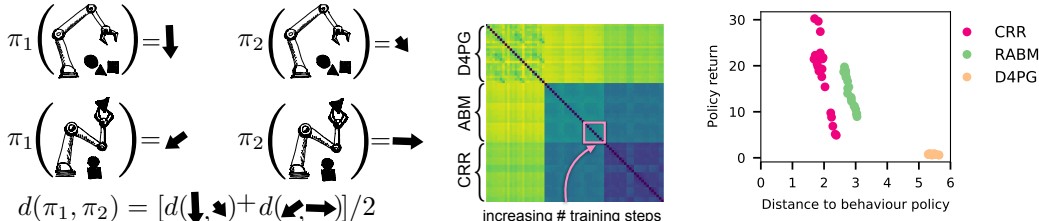

Figure 3: Left: The distance between two policies $\pi_1$ and $\pi_2$ is computed as an average distance across pairs of action vectors that are selected by each policy on a set of states. Middle: Pairwise distance matrix between policies on `humanoid-run` task. High distance is shown in yellow and low distance in dark blue. The $3 \times 3$ block structure of the matrix reflects the 3 types of training algorithm. Smaller blocks correspond to the increasing number of training steps with the same hyperparameters. Right: Policy value ($\mu_\pi$) versus action distance to the behavior policy $\bar{d}(\pi, \pi_\beta)$. Policies with the same training algorithm are in the same color.

A GP is a stochastic process, that is an indexed collection of random variables, such that every finite collection of them has a multivariate normal distribution [54]. In our case, we assume that the policy value $\mu$ as a function of policy $\pi$ follows a GP, with a kernel function $\mathcal{K}(\pi_i, \pi_j)$ over policies and mean $m$. Specifically, we consider the following generative process for the latent policy value $\mu(\pi)$, OPE estimate $\rho$, and episodic return $r$:

$$\mu(\pi) \sim \text{GP}(m(\cdot), \mathcal{K}(\cdot, \cdot)), \quad \rho \sim \mathcal{N}(\mu(\pi), \sigma_\rho^2), \quad r \sim \mathcal{N}(\mu(\pi), \sigma_r^2). \tag{3}$$

We use a constant mean $m$ without loss of generality. We assume a flat prior for the hyper-parameter $m$, and weakly informative inverse Gamma prior for the variance $\sigma_\rho^2$ and $\sigma_r^2$. Our probabilistic model is illustrated in Fig. 2.

As we have a finite number of policies, the input space is a finite set $\mathcal{X} = \{\pi_1, \ldots, \pi_K\}$, where each index corresponds to a candidate policy. The joint distribution of the $K$ policy values reduces to a multivariate normal distribution,

$$\mu_1, \ldots, \mu_K \sim \mathcal{N}(m\mathbf{1}_K, \mathbf{K}), \text{ with } \mathbf{K}_{k,k'} = \mathcal{K}(\pi_k, \pi_{k'}), \tag{4}$$

where $\mathbf{1}_K$ is a $K$-length vector of ones. For each policy $\pi_k$, we have one noisy observation from an OPE method $\rho_k$, and zero or multiple noisy episodic return observations $r_k^i$. As the policies are related through the covariance matrix $\mathbf{K}$, observing a return sample from one policy updates the posterior distribution of all of them. Given the OPE estimates $\boldsymbol{\rho} = [\rho_1, \ldots, \rho_K]$ and $N_k$ return observations $\mathbf{r}_k = [r_k^1, \ldots, r_k^{N_k}]$ for each policy $\pi_k$, the posterior distribution of the mean return is also a Gaussian,

$$\mu_1, \ldots, \mu_K | \boldsymbol{\rho}, \{\mathbf{r}_k\}_k \sim \mathcal{N}(\mathbf{m}, \boldsymbol{\Sigma}), \text{ with }$$
$$\mathbf{m} = \mathbf{K}(\mathbf{K} + \boldsymbol{\Lambda})^{-1}\mathbf{y}, \qquad \boldsymbol{\Sigma} = \mathbf{K} - \mathbf{K}(\mathbf{K} + \boldsymbol{\Lambda})^{-1}\mathbf{K}.$$
$$\mathbf{y}_k = \left(\frac{1}{\sigma_\rho^2} + \frac{N_k}{\sigma_r^2}\right)^{-1} \left(\frac{\rho_k}{\sigma_\rho^2} + \frac{1}{\sigma_r^2}\sum_{i=1}^{N_k} r_k^i\right), \qquad \boldsymbol{\Lambda}_{kk} = \left(\frac{1}{\sigma_\rho^2} + \frac{N_k}{\sigma_r^2}\right)^{-1}, \tag{5}$$

where $\boldsymbol{\Lambda}$ is a diagonal matrix with diagonal entries defined above.

## 2.3  Kernel

A key component of the GP model is the kernel that measures our belief about the policy correlation. To obtain a kernel we make a key assumption: *Policies that are similar in the actions that they take, yield similar returns*. Then, our insight is to measure the distance between the policies through the actions that each of them takes on a fixed set of states from the offline dataset (Fig. 3, left).

Formally, to measure the similarity between $\pi_1$ and $\pi_2$, we obtain the policy actions at a given state $s$ and measure the distance between these two action vectors $d(\pi_1(s), \pi_2(s))$ [3]. We then define the distance between the two policies as the average distance over the states of the offline dataset $\mathcal{D}$,

$$\bar{d}(\pi_1, \pi_2) = \mathbb{E}_{s \sim \mathcal{D}} \, d(\pi_1(s), \pi_2(s)). \tag{6}$$

---

[3]In this work, we use Euclidean distance for deterministic policies with continuous actions and Hamming distance for discrete actions. An extension to stochastic policies is straightforward.

In practice we approximate $\bar{d}$ using a randomly sampled subset of states $\mathcal{D}' \subset \mathcal{D}$.

An example of a pairwise distance matrix for a set of policies is shown in Fig. 3, middle. Rows and columns are the policies which are arranged first by the type of the training algorithm, then by hyperparameters and finally by the training steps (none of which is observed by our method). Note the block structure, which reflects the different nature of training algorithms. Smaller blocks of increasing policy distances correspond to policies with more training steps and the same hyperparameters.

Additionally, Fig. 3, right shows an example of policy distances to the behaviour policy. The values of policies produced by the same algorithm change gradually as a function of this distance. It provides indication that our kernel construction is informative for predicting the value of a policy.

Finally, we compute Matérn 1/2 kernel as: $\mathcal{K}(\pi_1, \pi_2) = \sigma_k^2 \exp(-\bar{d}(\pi_1, \pi_2)/l)$, where $\sigma$ and $l$ are the trainable variance and length-scale hyperparameters. This is a popular kernel choice [56]. As the distance metric is the average distance over states, using it in this kernel is equivalent to multiplication of kernels on these states and it is a valid positive semi-definite matrix.

### 2.4  Active offline policy selection with Bayesian optimization

Under a GP formulation, we can employ BO [56] to search for the best policy efficiently. BO optimizes an unknown function $\mu(\pi)$ with $\pi \in \mathcal{X}$ using a limited number of function queries that render noisy observations. BO has been used successfully in many applications including experimental design [5], robotics [42, 43], hyper-parameter optimization [59, 27, 12], preference learning and interactive machine learning [9, 8]. The key component of a BO algorithm is an acquisition function $u_i(\pi)$ that balances exploration and exploitation when selecting a query. Widely used acquisition functions include upper confidence bounds (UCB), expected improvement (EI) and epsilon-greedy. Here, we simply use the UCB acquisition function [60, 30].[4] Specifically, at every step $i$, we compute the score for a policy $\pi_k$ using

$$u_i(k) = \mathbf{m}_k + \sqrt{\beta_i \mathbf{\Sigma}_{kk}}, \tag{7}$$

where $\beta_i$ is a constant depending on $i$. We then choose the policy with the highest score, $k_i = \arg\max_k u_i(k)$, to execute next. After observing a new return $r_{k_i}$, we update $\mathbf{m}$ and $\mathbf{\Sigma}$ in Eq. 5. We also update the hyper-parameters ($m$, $\sigma_\rho^2$, $\sigma_r^2$, $\sigma_k^2$ and $l$) with the maximum a posteriori estimate (see details in the appendix).

Estimating the hyper-parameters with limited data is challenging in practice. Fortunately, in the active offline policy selection formulation, we can take advantage of the OPE estimates as prior observations to fit the hyper-parameters before launching the interactive optimization process. In some ablations where OPE is not available we instead execute 5 randomly sampled policies before fitting the hyperparameters.

## 3  Related work

Offline RL becomes increasingly popular due to its ability to leverage offline data to learn policies for RL in the real world. As a results, several benchmark datasets were recently announced [23, 19]. Offline RL methods include policy-constraint approaches that regularize the learned policy to stay close to the behavior policy [70, 21], value-based approaches that encourage more conservative estimates, either through regularization or uncertainty [35, 22], model-based approaches [73, 31], and standard RL methods used in off-policy RL [7].

OPE is studied extensively across many domains [39, 66, 29, 49]. It includes methods that use importance sampling [52] or stationary state distribution [41, 48, 67], value function [63, 44, 64], or learned transition models [74], as well as methods that combine these approaches [47, 15, 28, 18]. OPS is a related practical problem that receives increasing interest [50, 71, 20]. Our work has two notable distinctions in contrast to these prior works: 1) we consider similarities between policies when making value predictions, and 2) we focus on active setting, where a small number of evaluations in the environment can help to quickly improve the prediction of the best policy.

Similar to the way the kernel is computed based on policy's actions, Policy Evaluation Networks [24] work considers a network fingerprinting that learns a policy embedding through the outputs of the

---

[4]Experiments with other acquisition functions can be found in the supplementary materials.

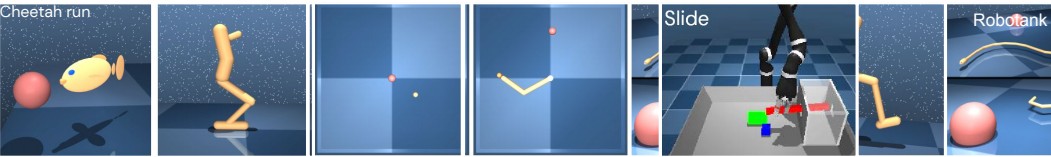

Figure 4: Environment used in the experiments. 9 environments from the dm-control domain (3 examples on the left), 4 from the MPG domain (2 examples in the middle), and 3 from Atari domain (1 example on the right).

policy on probing states. The principal difference to our approach is that we only consider a very limited online policy evaluation budget (in the order of $100$) while network fingerprinting needs many policy evaluations to train the policy embeddings (in the order of $100\,000$).

Applying BO in a finite set of indices reduces the problem to a Bayesian multiarmed bandit problem. Exploring the shared structure among arms using Bayesian models has been studied in the literature [60, 33, 26]. Our work builds upon the Bayesian bandit model to address the problem of policy selection additionally exploiting the offline dataset.

Active learning is used in RL for better exploration, learning reward or world models [17, 3, 14, 34, 6]. It has not been studied for the OPE or OPS problems to the best of our knowledge. Related to the policy representation through actions in the kernel, classifiers in active learning can be also characterised by their predictions on a subset of datapoints [32]. Furthermore, diversity in policies was measured through the action distribution for sampled states in population-based setting [51].

## 4 Experiments

### 4.1 Datasets and environments

In our experiments we consider three domains (Fig. 4) with different properties. Two domains are standard benchmarks in RL for continuous and discrete control and another one is motivated by the challenges of evaluating control policies in robotics. For each domain we describe the environments, datasets for offline RL and OPE, and sets of policies below. More details are provided in Appendix B.

**DeepMind Control Suite (dm-control)**   This is a standard set of continuous control environments [65]. It contains 9 tasks of locomotion and manipulation, some of which are shown in Fig. 4, left. The observations are the state of the MPD, including joint angles and velocities. The dimensionality and semantics of the continuous actions vary across tasks. We use the offline RL policies from DOPE [20] (a benchmark for OPE and OPS) that are trained using the dataset from the RL Unplugged benchmark [23]. For each task, there are up to 96 policies with various algorithms (D4PG [7], ABM [57], and CRR [70]), hyperparameters, and number of training steps.

**Manipulation Playground (MPG)**   This is a simulated robotics environment. The task is to control a Kinova Jaco arm with 9 degrees of freedom. Joint velocity control of 6 arm and 3 hand joints is used. The policies are learnt from proprioception and image input from frontal and in-hand cameras of $64 \times 64$ dimensions. We perform 4 manipulation tasks within a $20 \times 20$ cm basket (Fig. 4, middle). The dataset is generated by running an online RL algorithm as in [70] and the policies are generated as in [50]. For each task, there are up to 256 policies trained with various algorithms (including BC, D4PG [7], and CRR [70]), and hyperparameters.

**Atari**   This is a popular benchmark with discrete actions in online and offline RL [23]. For each of 3 games we trained 210 policies on an offline dataset [23] from high dimensional pixel observations with six offline RL methods, including Double DQN [68], CQL [35], BCQ [21], REM [2] and BVE [22]. As the range of returns is quite different from the other two domains, we scale the discounted returns to have approximately the same mean and variance based on the offline RL dataset.

We obtain OPE estimates for each policy by running Fitted Q-Evaluation (FQE) [44, 36] on the offline RL data. FQE is the default OPE method used in our experiments unless stated otherwise.

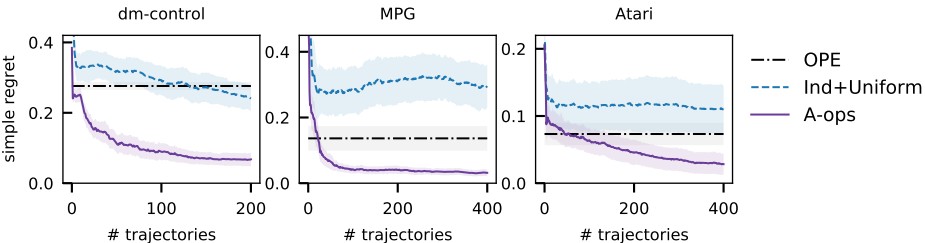

Figure 5: Comparison of A-OPS approach, offline policy selection with OPE, and online policy selection with IND+UNIFORM on dm-control, MPG and Atari environments. The results are averaged across 9, 4 and 3 tasks and 100 experiments in each of them. Shaded areas show standard deviation of the mean. Our method A-OPS quickly surpasses the baselines and achieves the lowest regrets.

## 4.2    Experimental setting and implementation

To evaluate the policy selection procedure in each experiment we randomly select a subset of $K$ policies out of all trained policies. We set $K = 50$ for dm-control and $K = 200$ for MPG and Atari. Then, we repeat each experiment 100 times, and report the average results and the standard deviation of the mean estimate. As a performance metric we monitor the simple regret (Eq. 2) as a function of the number of executed episodes. The algorithms are progressively evaluated when each trajectory is added. To make the results comparable across domains, we first compute the gap between the best and the worst performing policy in each environment across all the policies. Then, we normalise the scores in each experiment by dividing them by this gap. The results by environment are discussed in the appendix.

**Independent policy model**    To study the contribution of modeling the policy correlation with GP, we consider a baseline model with independently distributed value $\mu_k$ across policies. The data generating process for the $k$-th policy follows a hierarchical Gaussian distribution:

$$r_k^i \overset{iid}{\sim} \mathcal{N}(\mu_k, \sigma_{k,r}^2), \forall i, \text{ with } \mu_k \sim \mathcal{N}(\rho_k, \sigma_{k,\rho}^2), \tag{8}$$

where we assume a weakly informative inverse Gamma prior for variance $\sigma_{k,r}^2$ and $\sigma_{k,\rho}^2$ as in GP. We refer to this model as independent policy model (IND). Such model can be used in combination with any policy sampling strategy.

All online and offline RL algorithms, as well as offline policy evaluation algorithms in this work are implemented with Acme [25] and Reverb [11], and run on GPUs on an internal cluster. We implement GP and IND together with all BO algorithms using the same TensorFlow [1] codebase, and run all the policy selection experiments using CPUs. Details about the tasks and policies, hyperparameter values, and additional BO algorithms and ablations are presented in the appendix.

## 4.3    Comparison to the existing methods

In our main experiments we would like to verify that the proposed method performs better than two strategies commonly employed in practice (see section 1). To this end, we compare:

1. **A-OPS**: Active-Offline Policy Selection, our proposed method that uses OPE estimates to bootstrap GP learning and performs UCB policy sampling as discussed in section 2.
2. **OPE**: standard offline policy selection technique that selects the policy with the highest OPE estimate [50].
3. **IND+UNIFORM**: standard online selection where we execute policies uniformly and select the one with the highest return belief (estimated with independent policy model IND).

Fig. 5 presents the simple regret as a function of the number of executed trajectories. The results are averaged across 9 tasks from dm-control, 4 tasks from MPG and 3 games from Atari. Our proposed method A-OPS combines offline and online policy evaluations. It quickly outperforms OPE and improves with more trajectories. A-OPS estimate of the best policy may be erroneous initially due to noisy reward samples (*e.g.*, in MPG), but it quickly updates, thus allowing for significant improvements over OPE only after a few actively selected trajectories.

Table 1: Design of the ablation experiments across three axes: the use of OPE, GP or independent policy model, active policy selection or uniform sampling. We refer to our method that combines all the components as A-OPS.

| | **No OPE** | | **With OPE** | |
|---|---|---|---|---|
| | **Independent** | **GP** | **Independent** | **GP** |
| **Uniform** | IND+UNIFORM | GP+UNIFORM | IND+UNIFORM+OPE | GP+UNIFORM+OPE |
| **UCB** | IND+UCB | GP+UCB | IND+UCB+OPE | A-OPS |

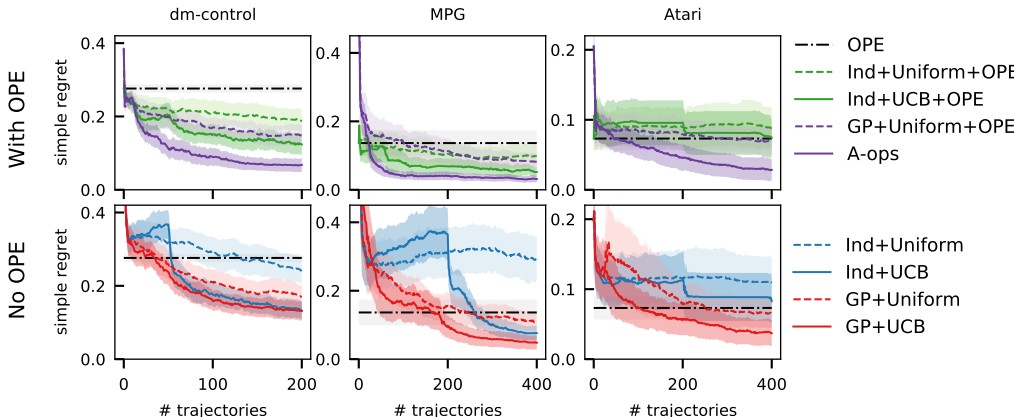

Figure 6: Ablations. In three domains, we ablate each of three components of our approach. 1) The use of OPE (with OPE in the first row and without in the second). 2) The choice of policy model: GP (purple and red) and independent (green and blue). 3) The policy selection strategy: active (dark, solid line) and uniform sampling (lighter, dashed line). In general, active is better than uniform, methods that exploit policy correlation are better than methods that do not, and methods that use an OPE prior are better than those that do not.

When comparing offline and online policy selection we notice that for the range of considered budgets, online policy selection IND+UNIFORM performs worse than OPE. It suggests that single trajectory returns are noisier than OPE estimates in these environments (more discussion on this is in the appendix). Naturally, we expect that given a larger budget the online baseline will eventually surpass its offline counterpart in all domains.

## 4.4 Component ablations

We are not aware of other works on active offline policy selection. Thus, in this section we investigate alternative solutions that help us to understand the importance of the proposed components of A-OPS. We design our ablation studies to test the contribution of the three components of A-OPS:

1) *OPE*: initialisation of algorithms with or without OPE estimates.
2) *active learning*: active policy selection (UCB) or uniform sampling (UNIFORM).
3) *policy kernel*: GP model or independent model baseline (IND).

Tab. 1 summarises the 8 possible combinations of the components. We refer to our full method (GP+UCB+OPE) simply as A-OPS. We also include the pure offline OPE estimates as before.

The results with all the baselines in two domains are presented in Fig. 6. First of all we observe that A-OPS outperforms all other methods across a wide range of budgets, thus, all the components are crucial for the performance of our method and excluding any of them leads to significantly worse results. Next, we answer three questions about each component individually.

**1. How important is it to incorporate OPE estimates?** Across all the experiments, incorporating OPE estimates (top row of Fig. 6) always considerably improves the results compared to starting policy selection from scratch (bottom row). For example, in MPG when A-OPS starts with OPE, it is able to cut the OPE regret by more than half before executing 100 trajectories. For this budget, GP+UCB, which starts from scratch, does not reach the same regret as OPE.

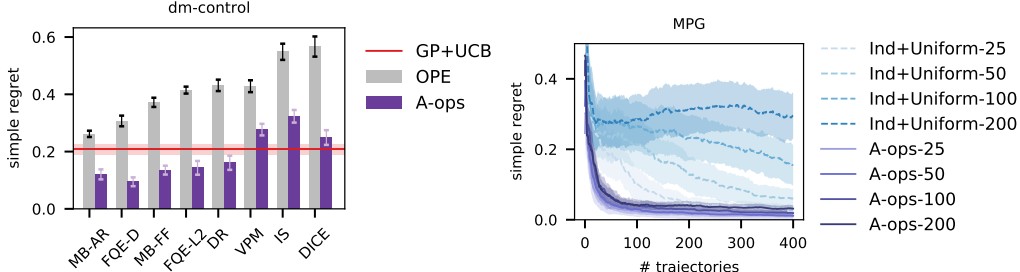

Figure 7: Left: The performance of A-OPS on dm-control with varying type of OPE estimate with a budget of 50 trajectories. The various OPE methods are Model-Based OPE with an auto-regressive transition model (MB-AR), Fitted Q Evaluation with distributional Q function and distributional loss (FQE-D), Model-Based OPE with a feedforward transition model (MB-FF), Fitted Q Evaluation with standard Q function and L2 loss (FQE-L2), Double Robust (DR), Variational Power Method (VPM), Importance Sampling (IS), Distribution Correction Estimation (DICE). A-OPS can improve upon the initial OPE estimates in all cases. Right: Performance of A-OPS and IND+UNIFORM with varying number of policies available for execution. IND+UNIFORM is affected a lot by the increase in the number of policies. Our approach A-OPS performs similarly when the number of policies is increase by a factor of almost 10.

**2. How informative is our kernel?**    To understand the value of modelling the correlation between policies through the kernel, we compare the GP method with the independent policy model (IND). In Fig. 6 purple and red lines use GP as the policy model and green and blue use the IND model. In the vast majority of settings (11 out of 12) the use of a kernel significantly improves the results and in one setting (IND+UNIFORM+OPE vs GP+UNIFORM+OPE in MPG domain) they perform on par. We conclude that the kernel is a key ingredient for improving data efficiency.

**3. How important is the selection strategy?**    The active selection of policies is generally beneficial for identifying a good policy. For this we refer to the results of dashed lines that correspond to UNIFORM policy selection and solid lines that correspond to the use of UCB. Using IND+UCB yields a high regret in the initial exploration stage, but it improves substantially over IND+UNIFORM later. Moreover, incorporating OPE estimates and a kernel (resulting in A-OPS) significantly shortens the exploration stage.

## 4.5    More ablation studies

Incorporating the OPE estimates is clearly beneficial for policy selection. Here, we further study: **How does the performance depend on the quality of OPE estimates?** Fig. 7, left compares A-OPS using estimates from different OPE algorithms in DOPE [20] on dm-control with a small budget of 50 trajectories. A-OPS always improves upon the corresponding OPE estimate, and the performance clearly depends on the quality of the associated OPE estimate. The performance of GP+UCB with 50 trajectories that does not rely on any OPE metric is shown as a horizontal line. Some OPE estimates (e.g., IS, VPM and DICE) have large estimation errors and even a negative correlation with the ranks of policies [20]. As expected, A-OPS is negatively impacted by poor OPE estimates and ignoring them could yield better results. It should be possible to correct for this to a greater extent with more online observations. For now we follow the recommendation to use the FQE estimates.

In subsection 2.2 we hypothesised that A-OPS method should be data efficient as it would scale with the diversity of policies and not their number. We next try to verify it and answer the questions **How does the policy selection algorithm scale with the growing number of candidate policies?** In Fig. 7, right we show the simple regret for A-OPS and IND+UNIFORM in the MPG domain when we sample 25, 50, 100 and 200 policies. We see that when the number of policies is not very high (25), the algorithm that treats policies independently and samples them at uniform (IND+UNIFORM-25) can closely approach the performance of our method A-OPS-25 after 100 trajectories. However, when the number of policies grows, this small interaction budget is not enough for the naive algorithm: IND+UNIFORM-200 significantly degrades its performance. At the same time, A-OPS-200 is minimally affected by the increase in the number of policies as the diversity of the policies stays approximately the same.

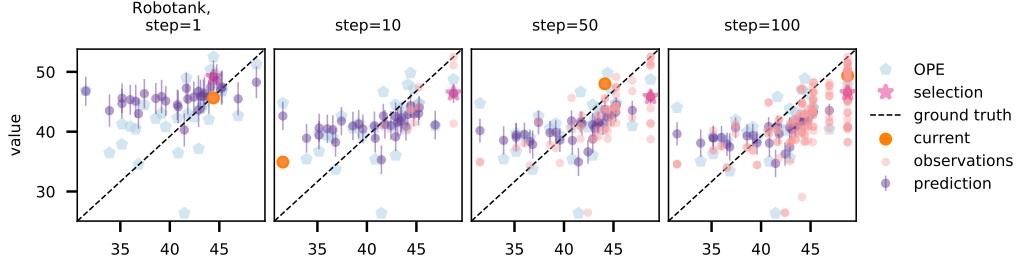

Figure 8: Visualisation of the qualitative performance of A-OPS algorithm on `Robotank` environment with 20 randomly selected policies (ordered by ground truth only for the illustration purpose). From left to right we show the state at 1, 10, 50, 100 algorithm iterations. The initial OPE scores are shown in light blue, the policy selected by A-OPS is highlighted with magenta star, the observation of the currently executed policy is shown in orange circle, past observations are in pink and the prediction with its standard deviation is in purple.

## 4.6 Qualitative results

Finally, to understand the behaviour of A-OPS method we show its progress for one of the experiments with Robotank game in Fig. 8. For visualisation purposes, we use only 20 policies which are ordered by their ground truth returns (not observed by the method). Perfect value predictions would be aligned along the diagonal line. We show the initial *OPE* values, current *selection* of the best policy, *current* observation of the executed policy, past *observations* and the algorithm *prediction* with its standard deviation for steps 1, 10, 50, and 100. The A-OPS prediction is determined by the OPE observations and past observations of the policy returns. The kernel reinforces smoothness of the function. We can see that the initial selection is dominated by OPE prediction and the selected policy is quite far from the best policy. With more online observations, the predictions improve. Our algorithm is data-efficient because 1) it samples more promising policies more frequently (more observations with higher scores), 2) the knowledge from one policy is propagated to the policies related by the kernel (predictions for some policies are updated without any new observations of their values).

## 5 Conclusions

In this paper, we introduced the active offline policy selection problem. We proposed a BO solution that integrates OPE estimates with evaluations obtained by interacting with the environment. We also designed a novel kernel that models correlation over policies. Experiments on three control domains proved that our solution is very effective. Furthermore, our experiments ablated the different components of the solution, highlighting the value of modelling correlation via the kernel, active selection, and incorporation of OPE estimates in the observation model.

**Limitations and future work**    Having tested the method and demonstrated its efficacy in simulation, our future plan is to conduct experiments in real domains.

The A-OPS approach could be of use in many safety critical applications. This work does not deal with this question explicitly as we assume safety constraints to be implemented directly on the hardware. It means that if an unsafe policy is attempted, the safety controller would terminate the policy before reaching the end of an episode, resulting in low reward and thus discouraging A-OPS from trying this policy again. Alternatively, we could alleviate this limitation by applying constrained BO techniques [62] to search for the best policy without violating safety constraints.

In this work, we explore the policy structure based on the offline dataset, and use every new trajectory only to obtain a noisy observation of policy values. This allows a clean Bayesian optimization formulation. Going forward, more information could be extracted from the new trajectories, for example, to further improve OPE estimates, GP kernel, or policies themselves. However, using this information would make the solution much more complex and we may need to take the additional computational cost into account. Finally, to construct a better policy kernel, we may investigate how to measure the policy similarity on the most informative subset of states in the offline dataset.

## Acknowledgements

We would like to thank Bobak Shahriari for the useful discussions and feedback on the paper.

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
