# A Algorithmic details

## A.1 GP model

Following the data generative process in Sec. 2.2, the hierarchical probabilistic model of all the latent policy values $\mu$, OPE estimates $\rho$, and episodic returns $r$[5] at the given $K$ policies $\{\pi_k\}_{k=1}^K$ is given as follows:

$$
\begin{aligned}
\mu_1, \ldots, \mu_K &\sim \mathcal{N}(m\mathbf{1}_K, \mathbf{K}) \\
\rho_k &\overset{i.i.d.}{\sim} \mathcal{N}(\mu_k, \sigma_\rho^2), \forall k \in \{1, \ldots, K\} \\
r_k^i &\overset{i.i.d.}{\sim} \mathcal{N}(\mu_k, \sigma_r^2), \forall i \in \{1, \ldots, N_k\} \\
\sigma_\rho^2 &\sim \text{IG}(\alpha, \beta) \\
\sigma_r^2 &\sim \text{IG}(\alpha, \beta),
\end{aligned}
\tag{9}
$$

where the mean $m$ is a constant hyper-parameter and the covariance $\mathbf{K}(\pi_k, \pi_k') = \mathcal{K}(\pi_k, \pi_{k'})$. The kernel function is defined as

$$
\begin{aligned}
\mathcal{K}(\pi_k, \pi_{k'}) &= \sigma_k^2 \exp(-\bar{d}(\pi_k, \pi_{k'})/l) \\
\bar{d}(\pi_k, \pi_{k'}) &= \mathbb{E}_{s \sim \mathcal{D}'} \, d(\pi_k(s), \pi_{k'}(s)) \\
\sigma_k^2 &\sim \text{IG}(\alpha, \beta),
\end{aligned}
\tag{10}
$$

with $\mathcal{D}'$ being a fixed set of states that is randomly subsampled from the entire offline dataset $\mathcal{D}$. $\sigma_k^2$ and $l$ are hyper-parameters. We assume an uninformative prior for $l$.

The posterior distribution of $[\mu_1, \ldots, \mu_K]$ given all the observation and hyper-parameters is also a multivariate normal distribution given in Eq. 5. The log-marginal likelihood of all the hyper-parameters with $\mu_k$'s integrated out is as follows,

$$
\begin{aligned}
&\log P(\boldsymbol{\rho}, \{\mathbf{r}_k\}_k, \sigma_\rho^2, \sigma_r^2, \sigma_k^2 | m, l) \\
&= -\frac{1}{2}\log|\boldsymbol{\Sigma}| - \frac{1}{2}\mathbf{y}^T\boldsymbol{\Sigma}\mathbf{y} - \frac{1}{2}\left(K\log(\sigma_\rho^2) + n\log(\sigma_r^2) - \sum_{k=1}^K \log(\boldsymbol{\Lambda}_{kk})\right) \\
&\quad -\frac{1}{2}\sum_{k=1}^K\left(\frac{\rho_k^2}{\sigma_\rho^2} + \frac{1}{\sigma_r^2}\sum_{i=1}^{N_k}(r_k^i)^2 - \frac{\mathbf{y}_k^2}{\boldsymbol{\Lambda}_{kk}}\right) + \log \text{IG}(\sigma_\rho^2) + \log \text{IG}(\sigma_r^2) + \log \text{IG}(\sigma_k^2) + \text{const.},
\end{aligned}
\tag{11}
$$

where $n = \sum_k N_k$ and const. is a constant.

## A.2 Independent policy value baseline model (IND)

In Sec. 4.2, we consider a baseline model with independent policy value distributions as follows,

$$
\begin{aligned}
r_k^i &\overset{i.i.d.}{\sim} \mathcal{N}(\mu_k, \sigma_{k,r}^2), \forall i, \text{ with} \\
\mu_k &\overset{i.i.d.}{\sim} \mathcal{N}(\rho_k, \sigma_{k,\rho}^2), \forall k \\
\sigma_{k,\rho}^2 &\overset{i.i.d.}{\sim} \text{IG}(\alpha, \beta), \forall k \\
\sigma_{k,r}^2 &\overset{i.i.d.}{\sim} \text{IG}(\alpha, \beta), \forall k.
\end{aligned}
\tag{12}
$$

---

[5]We assume Gaussian distribution for the noise model, but if the form of return noise distribution is known in advance, it can replace the Gaussian noise model and approximate GP inference can be performed.

Since the model is i.i.d. across policies, we omit the index $k$ for ease of notation. The posterior distribution of $\mu$ given $\mathbf{r} = [r^1, \dots, r^N]$, $\sigma_r^2$ and $\sigma_\rho^2$ is then given as

$$\mu \sim \mathcal{N}(m, s^2)$$
$$m = \frac{\sum_{i=1}^N r^i + (\sigma_r^2/\sigma_\rho^2)\rho}{N + (\sigma_r^2/\sigma_\rho^2)}$$
$$s^2 = \frac{\sigma_r^2}{N + (\sigma_r^2/\sigma_\rho^2)} \,. \tag{13}$$

The log-marginal likelihood with $\mu$ integrated out can be derived as

$$\log P(\rho, \mathbf{r}, \sigma_\rho^2, \sigma_r^2|)$$
$$= -\frac{1}{2}\left(\log(\sigma_\rho^2) + N\log(\sigma_r^2) - \log(s^2)\right) - \frac{1}{2}\left(\frac{\rho^2}{\sigma_\rho^2} + \frac{1}{\sigma_r^2}\sum_{i=1}^N (r^i)^2 - \frac{y^2}{s^2}\right)$$
$$+ \log \mathrm{IG}(\sigma_\rho^2) + \log \mathrm{IG}(\sigma_r^2) + \mathrm{const.} \,, \tag{14}$$

where const. is a constant.

## A.3 Hyperparameter fitting

We fit the hyperparameters in both models whenever a new observation is obtained by searching for the maximum a posteriori (MAP) estimate using a few steps of gradient descent on the log-marginal likelihood.

Fitting the hyper-parameters at the beginning with few observations is challenging. When we do not use OPE to warm start GP inference such as in GP+UCB and GP+UNIFORM, we start by randomly selecting the first 5 policies to execute. Besides, it is important in this case to apply a prior for hyperparameters to prevent variance from shrinking towards zero. When OPE is available, we find neither of these techniques is required for a robust estimation and we can start fitting the hyperparameters of GP from the first step.

## A.4 Hyperparameter values

The initial values of the hyperparameters of GP and our method in general are listed next. We set the GP hyperparameters based on the statistics about the policy returns from the offline dataset. The same hyperparameters are used for all the domains. The discounted returns in Atari domain are very different in scale from the other domains. To ensure that the same hyperparameters are applicable to all domains, we rescale returns in Atari to the same range of means and variances as other domains based on the statistics from the datasets for offline RL.

- We find using a constant value $\sqrt{\beta_i} = 5$ in GP-UCB works well across different budgets, Bayesian models, and environments, although a more sophisticated scheduling [61, 4, 26] could further improve the results.

- The hyperparameters of the prior for IND were set to: $\alpha = 1, \beta = 1000$.

- Observation noise variance of GP was set to the initial value of 1000 or prior with $\alpha = 1, \beta = 200$ when the prior is used. Initial value of the offset $m$ was 0.

- For GP optimisation we used Adam optimiser with learning rate 0.001 and 1000 optimisation steps per observation update. $\beta_1 = 0.9, \beta_2 = 0.999$.

- To compensate to the non-zero mean in the GP, we use a constant component in the kernel with trainable variance parameter initially set to 10. When the prior is used, we set $\alpha = 1, \beta = 200$ in all the experiments.

## A.5 Kernel selection

In our preliminary experiments, we experimented with a few ways to compute the kernel over policies. To quickly select promising ideas, we tested them in a simple regression task where we split all

policies with their true returns into training and testing sets and predict the returns of the test policies based on the training policies. Then, we select the kernel that gives the highest log likelihood and lowest squared prediction error. We choose the best type of kernel using 3 dm-control environments (`cartpole_swingup`, `fish_swim` and `manipulator_insert_ball`) and apply the same kernel to all tasks of both domains in the experiments. The most important variations of the kernel that we studied are enumerated below. We highlight the variant that was used in the final experiments.

- As a subset $\mathcal{D}'$ of states for computing policy's actions, we select $1000$ states. We tried to use 1) *random states*, 2) initial states, 3) states with high rewards, 4) states with low rewards, 5) states in the trajectories with high returns, 6) states in the trajectories with low returns. Several options looked promising, including 1, 2, 4. At the end we chose the simplest option that yielded good results (1).

- As a way to construct a kernel from the policy actions on several states, we considered: 1) *multiplication of kernels on separate states*, 2) concatenating action predictions into a single feature vector. We chose kernel multiplication because we found in our experiments that concatenating the action in a single vector makes the distances not very informative. This happens because the dimensionality of the policy representation becomes very high.

- After obtaining the distances matrix for the policies, we compute the kernel. We tried 1) RBF, 2) Matérn $5/2$, 3) Matérn $3/2$, 4) *Matérn $1/2$* kernels. While the difference between these versions was not large, the less smooth kernel Matérn $1/2$ performed slightly better and reflected our intuition that the return of policies trained with different algorithms can change quickly (Fig. 3, right).

### A.6 Bayesian optimization algorithm

The pseudo-code of the Bayesian optimization algorithm for both models is presented in algorithm 1.

---

**Algorithm 1:** UCB algorithm for active offline policy selection

**Data:** Environment env, set of policies $\{\pi_k\}$, $1 \leq k \leq K$, number of initial random samples $n_{\text{init}}$, total number of steps $n$, (optional) OPE estimates $\{\rho_k\}$

**Result:** Final recommendation $\hat{k}$

**for** $i : 1 \rightarrow n$ **do**
    **if** $i \leq n_{\text{init}}$ **then**
        Sample a policy $k_i$ uniformly from $\{1, 2, \ldots, K\}$
    **else**
        Fit hyper-parameters by maximizing the log-marginal likelihood (Eq. 11 or Eq. 14) with all the return samples $r_i$ and OPE estimates $\rho_k$;
        Update the posterior of $\boldsymbol{\mu}$ using Eq. 5 or Eq. 13;
        Compute the UCB acquisition function $u_i(k)$ for every policy $k$ using Eq. 7;
        Choose a policy $k_i = \arg\max_k u_i(k)$;
    **end**
    Apply policy $\pi_{k_i}$ in env for one episode and receive the discounted episodic return $r_i$;
**end**
Maximize the log-marginal likelihood (Eq. 11 or Eq. 14) wrt the hyper-parameters using Adam;
Update the posterior of $\boldsymbol{\mu}$ using Eq. 5 or Eq. 13;
**return** $\hat{k} = \arg\max_k \mathbf{m}_k$ *with* $\mathbf{m}_k$ *being the posterior mean of $k$-th policy*

---

## B Domains details

In this section, we present the statistics about our domains of interest: dm-control, MPG and Atari. We describe how we obtain the policies and present the statistics about the returns.

### B.1 Dm-control

In dm-control domain we use policies from the DOPE benchmark [20], which results in a diverse set of policies with varying algorithms (D4PG, ABM, CRR), hyperparameters like learn-

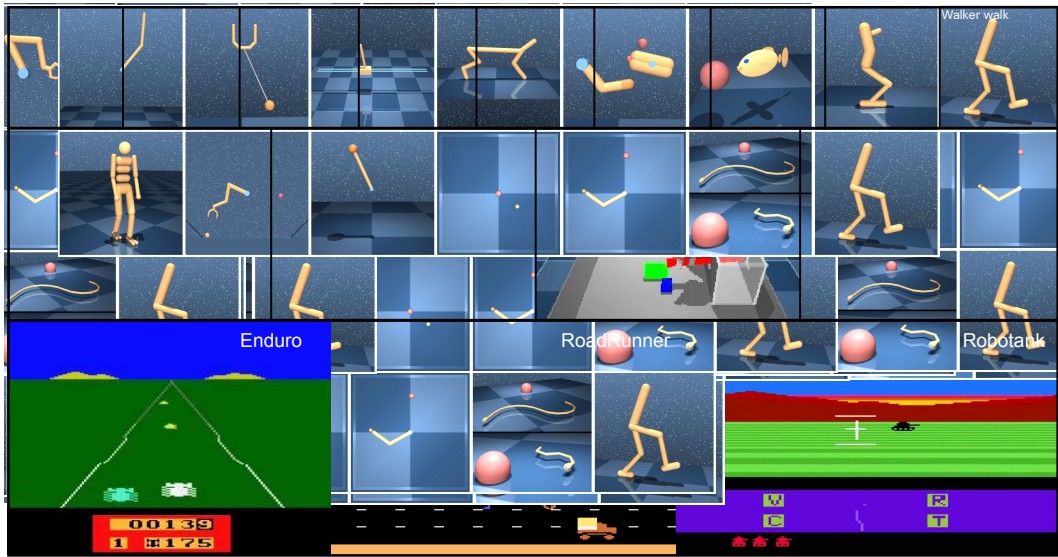

Figure 9: Environments used in the experiments. Top: 9 environments from the dm-control domain. Middle: 4 environments from the MPG domain. Bottom: 3 game environments from Atari domain.

Table 2: Statistics about the policies in dm-control: the return of the worst policy, of the best policy and standard deviation of the policies returns.

| task | stdev | min return | max return |
|---|---|---|---|
| cartpole_swingup | 6.488 | 0.451 | 31.504 |
| cheetah_run | 5.504 | 6.933 | 36.291 |
| finger_turn_hard | 14.055 | 8.613 | 60.971 |
| fish_swim | 3.208 | 8.097 | 18.755 |
| humanoid_run | 8.993 | 0.566 | 30.117 |
| manipulator_insert_ball | 2.825 | 3.668 | 13.801 |
| manipulator_insert_peg | 2.306 | 4.668 | 11.249 |
| walker_stand | 13.173 | 39.180 | 85.533 |
| walker_walk | 13.973 | 11.950 | 74.719 |

ing rate, and number of training steps, giving up to 96 policies per task, and a diverse range of returns. There are 9 environments as shown in the top row of Fig. 9: cartpole_swingup, cheetah_run, finger_turn_hard, fish_swim, humanoid_run, manipulator_insert_ball, manipulator_insert_peg, walker_stand, walker_walk. There is a significant gap between the scores of the badly-performing and well-performing policies. Tab. 2 shows the standard deviation, minimum and maximum returns of the policies from the dataset.

The success of the active policy selection strategy in each experiment depends on the relation between the variance of the returns of the policies in the dataset and the variance of the returns that a single policy renders. To understand this relationship better in Fig. 10 we show for each environment: 1) the distribution of policy OPE metrics as a function of true scores at the top, and 2) the histogram of the episodic returns for a single randomly selected policy (highlighted and connected by a dashed line) at the bottom. We observe that in some cases (for example, in finger_turn_hard, fish_swim, walker_walk) the variance of episodic returns is greater than the variance of the policies true scores. This is particularly challenging for any policy selection that involves the online component. Furthermore, we also observe that in some cases OPE scores are not well correlated with the true scores (for example, fish_swim, humanoid_run, manipulator_insert_ball, manipulator_insert_peg). In this case, on one hand, relying on OPE scores makes policy selection more challenging, but on the other hand, it leaves the biggest potential for improvement by involving the online evaluation.

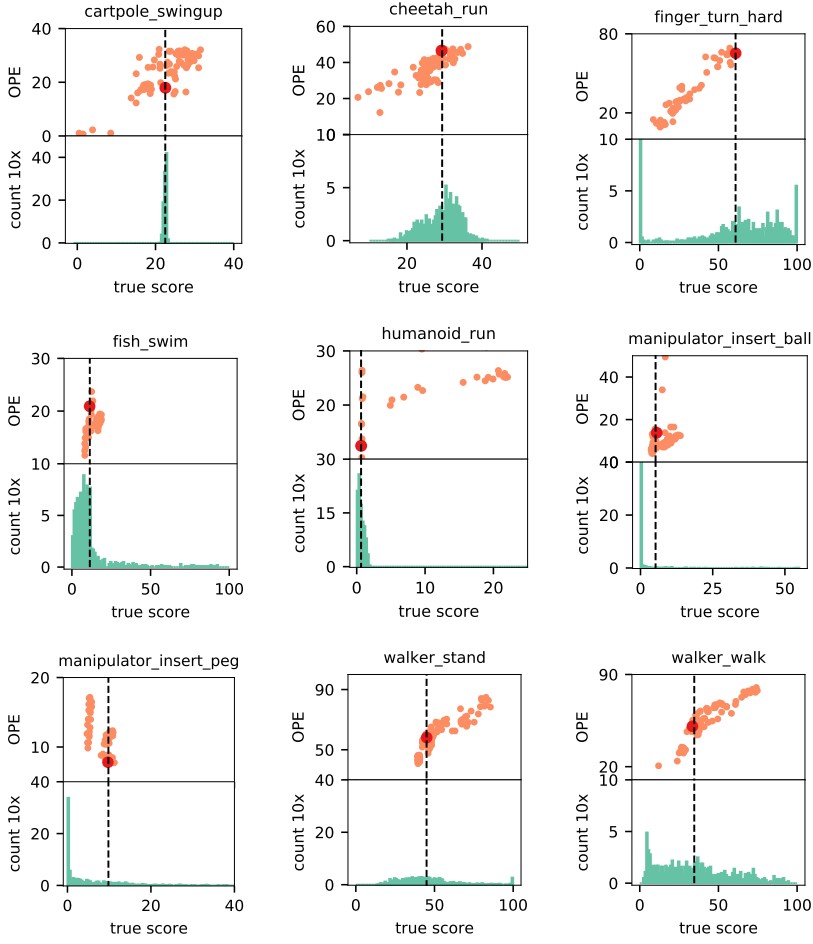

Figure 10: For each environment in dm-control we show 1) the distribution of policy OPE metrics as a function of true scores at the top (in orange), and 2) the histogram of the episodic returns for a single randomly selected policy (in green) at the bottom. The randomly selected policy is highlighted with a red dot on the top and it is connected by a dashed line to the mean values in the histogram.

## B.2 MPG

To generate policies for the environments in MPG domain we follow the protocol in [50]. For each task, we generate up to 256 policies trained with various algorithms (BC, D4PG, and CRR), and hyperparameters including number of hidden units (64, 1024), number of blocks (1, 5), learning rate ($1 \times 10^{-3}$, $1 \times 10^{-5}$), learning steps (50k, 250k, 25k), and for CRR the beta parameter (0.1, 10). This ensures there is a diverse range of policies that span the range of possible returns. There are four tasks in MPG domain: `box`, `insertion`, `slide`, and `stack_banana` as shown in the middle row of Fig. 9. We include MPG as a challenging domain where policies are learnt from high dimensional images and training online policy takes thousands of trajectories (around 8000) even when human demonstrations are included.

The policies in this domain also exhibit a significant gap and large variance in the scores of the badly-performing and well-performing policies as it is demonstrated by the statistics in Tab. 3. As this is a challenging domain, in all the environments, the worst policy achieves 0 (lowest possible) regret.

In Fig. 11 we study again the relationship between the distribution of policy OPE metrics and true scores, as well as the distribution of the episodic returns for a random policy. In all environments, the variance of episodic returns is comparable or even greater than the variance of the policies true scores.

Table 3: Statistics about the policies in MPG: the return of the worst policy, of the best policy and standard deviation of the policies returns.

| task | stdev | min return | max return |
|------|-------|-----------|-----------|
| box | 19.711 | 0.0 | 64.58 |
| insertion | 15.808 | 0.0 | 52.42 |
| slide | 14.622 | 0.0 | 52.17 |
| stack_banana | 18.420 | 0.0 | 58.74 |

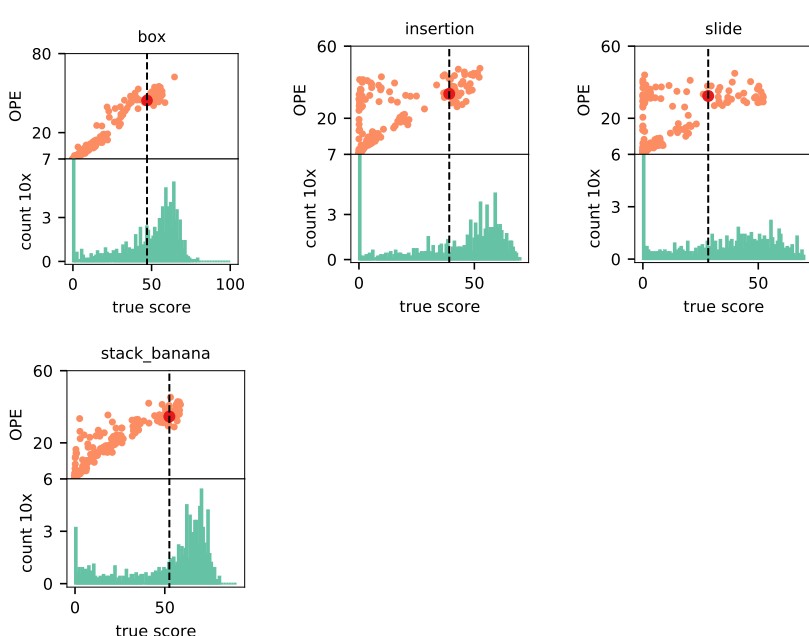

Figure 11: For each environment in MPG we show 1) the distribution of policy OPE metrics as a function of true scores at the top (in orange), and 2) the histogram of the episodic returns for a single randomly selected policy (in green) at the bottom. The randomly selected policy is highlighted with a red dot on the top and it is connected by a dashed line to the mean values in the histogram.

In some cases, like in `box` and `stack_banana` the OPE scores are almost perfectly correlated with the true returns, which makes the OPE baseline very hard to beat. Even when OPE scores are not perfectly correlated with the true returns, like in the `insertion` environment, it may still be very informative for the task of selecting the *best* policy. There is the biggest potential for improvement over OPE in `slide` environment. Also, notice that many policies have a peak value in the histogram at value 0 (including policies with high average return). This property makes the sparse online evaluation particularly challenging.

## B.3 Atari

We choose Atari in order to test our method on an established discrete control offline RL domain where training an online policy from pixels is challenging and requires thousands of trajectories. We follow the protocol in [23] to train 210 policies with different architectures and hyperparameter for each of three Atari games: `Enduro`, `RoadRunner`, and `Robotank` as shown in Fig. 9, bottom row. `Enduro` is previously used as an OPE benchmark [69]. We include `RoadRunner` and `Robotank` to cover a range of difficulty and reward scales presented in the Atari benchmark. The policies cover a diverse range of methods and this results in the most diverse set of policies for the experiments in this paper: DDQN [68], BCQ [21], REM [2], IQN [13], CQL [35], and BVE [22]. We use grid search over the following hyperparameters: learning rate for all algorithm ($3 \times 10^{-5}$, $1 \times 10^{-4}$, $3 \times 10^{-4}$), the pessimism regularizer in CQL ($1 \times 10^{-3}$, $1 \times 10^{-2}$, $1 \times 10^{-1}$, 1, 2), the threshold parameter in

Table 4: Statistics about the policies in Atari: the return of the worst policy, of the best policy and standard deviation of the policies returns.

| task | stdev | min return | max return |
|---|---|---|---|
| Enduro | 3.198 | 4.848 | 25.617 |
| Robotank | 5.509 | 9.786 | 49.898 |
| RoadRunner | 3.867 | 1.012 | 26.298 |

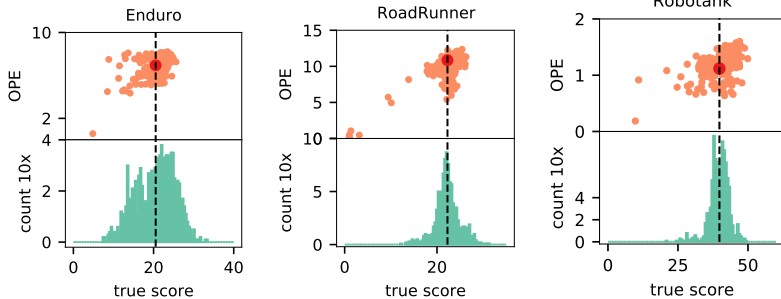

Figure 12: For each environment in Atari we show 1) the distribution of policy OPE metrics as a function of true scores at the top (in orange), and 2) the histogram of the episodic returns for a single randomly selected policy (in green) at the bottom. The randomly selected policy is highlighted with a red dot on the top and it is connected by a dashed line to the mean values in the histogram.

BCQ (0.1, 0.25, 0.5, 0.75). Each algorithm with a hyperparameter setting is trained on 5 datasets generated by DQN snapshots trained with five different seeds as described in [2].

As in other domains, there is a significant gap between the scores of the badly-performing and well-performing policies as indicated in the Tab. 4. In Fig. 12 we show the relationship between the distribution of policy OPE metrics and true scores, as well as the distribution of the episodic returns for one of the policies. The OPE scores are correlated, but the correlation is far from perfect and there is space for improvement by including the online trajectories.

## C Results by task

While subsection 4.3 and subsection 4.4 present the aggregated results across all tasks where each task is normalised by the gap between policies, in this section we present figures for each task separately. We discuss the variability of the results across different environments and explain the reasons for this variability.

### C.1 Dm-control

Similar to Fig. 5, Fig. 13 shows the comparison between our proposed method A-OPS, completely offline policy selection with OPE and completely online selection with IND+UNIFORM on each of 9 task from dm-control. Depending on the quality of the initial OPE values and the variance of the policy returns (see Fig. 10), A-OPS may take different number of trajectories before it outperforms all the baselines, but usually it only takes a few steps. Eventually, A-OPS reaches the best performance with the limited budget.

Similar to Fig. 6 (first row), Fig. 14 shows the contribution of each of the components of the method. Our method A-OPS is preferable in all environments across a wide range of interaction budget except for `cheetah_run` with less than 50 trajectories. Again we observe that modelling correlated policies as in GP performs better than modelling independent policies as in IND, active policy selection as in UCB is better than uniform policy selection as in UNIFORM. In manipulator tasks, no method achieves a regret as low as in other tasks. We believe that the main reasons for this are 1) low performance of the initial OPE estimates and 2) the skewed distribution of episodic returns of all policies where most returns are close to 0 (see Fig. 10).

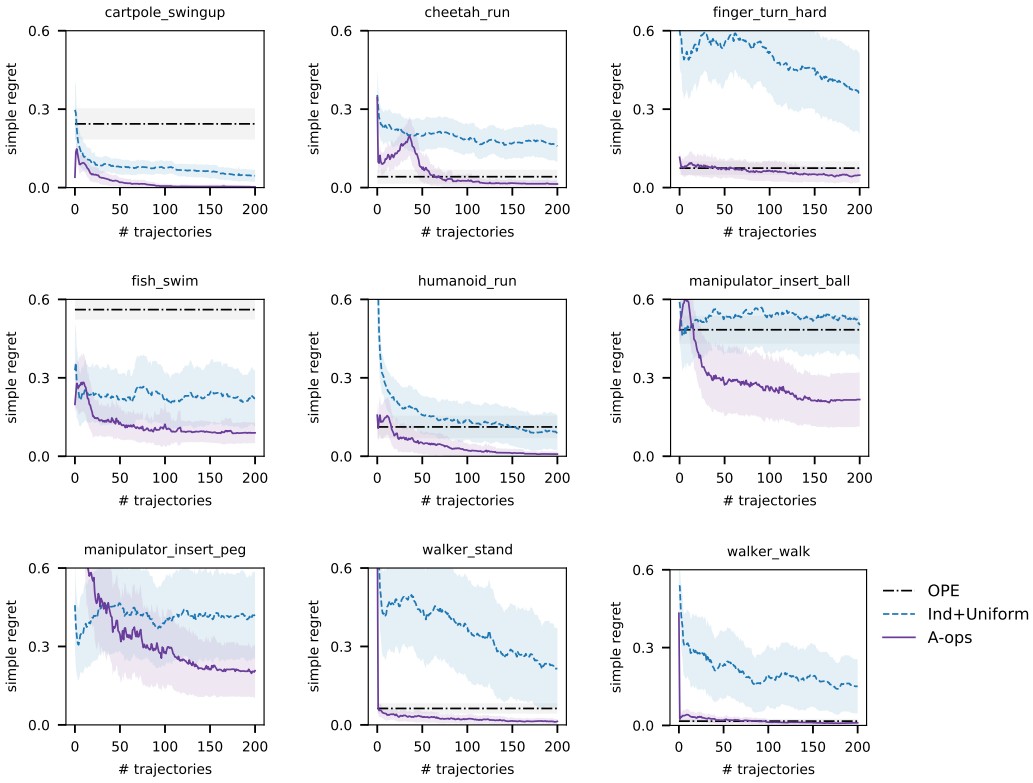

Figure 13: Comparison of A-OPS approach, offline policy selection with OPE, and online policy selection with IND+UNIFORM on each environment in dm-control domain. The results are averaged across 100 experiments. We do not show OPE performance in `manipulator_insert_peg` in order to respect the y-axis limits. The value is 0.88 as D4PG policies are highly overestimated. Shaded areas show standard deviation of the mean. Our method A-OPS quickly surpasses the baselines and achieves the lowest regrets.

Similar to Fig. 6 (second row), Fig. 15 shows the contribution of each of the components of the method in case when OPE is not used. It is clear that the results are significantly worse than when using OPE (Fig. 14) which clearly indicated the benefit of OPE component in A-OPS. When OPE estimates are not available, the combination of modelling correlated policies as in GP and intelligent policy selection as in UCB gives the best results on average. It performs better than the next best method in 6 environments, slightly worse in 1 and approximately the same in 2. On average, the GP+UCB strategy is the best when OPE estimates are not available.

## C.2 MPG

Similar to Fig. 5, Fig. 16 shows the comparison between our proposed method A-OPS, completely offline policy selection with OPE and completely online selection with IND+UNIFORM on each of 4 tasks from MPG. OPE performs exceedingly well in 3 of 4 tasks, getting regret close to zero for 2 tasks (see Fig. 11 with plot of OPE vs the ground truth). Nevertheless we manage to perform about as well or better on all of the tasks: In 2 environments, A-OPS only approaches the OPE baseline, but in the other 2 environments, A-OPS quickly surpasses OPE. It makes the most improvement in the `slide` task. This difference in the performance in different tasks is due to the variability of performance of OPE method and the variance in returns in the online policy executions. High return variance requires a large number of environment interactions for an online policy evaluation method to provide accurate estimates. However, the most important observation is that A-OPS achieves a small regret in all environments.

Similar to Fig. 6 (first row), Fig. 17 shows the contribution of each of the components of the method when using OPE. Our method A-OPS is preferable in all environments across a wide range of interaction budgets. The same observation as before holds: modelling correlated policies as in GP

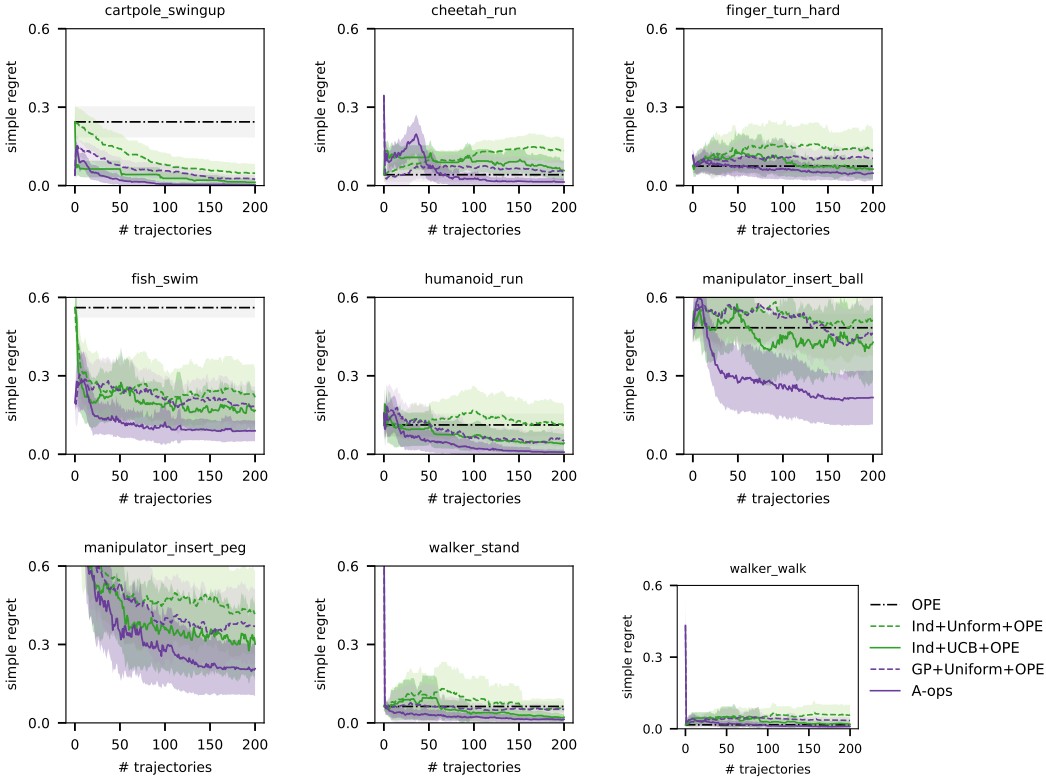

Figure 14: Ablations. In 9 environments of dm-control domain, we ablate two components of our approach when using OPE metrics. 1) The choice of policy model: GP (purple) and independent (green). 2) The policy selection strategy: active (dark, solid line) and uniform sampling (bright, dashed line). We do not show OPE performance in `manipulator_insert_peg` in order to respect the y-axis limits. The value is $0.88$ as D4PG policies are highly overestimated. In general, active is better than uniform, methods that exploit policy correlation are better than methods that do not.

performs better than modelling independent policies as in IND, active policy selection as in UCB is better than uniform policy selection as in UNIFORM.

Similar to Fig. 6 (second row), Fig. 18 shows the contribution of each of the components of the method in case when OPE is not used. As before, the results are significantly worse than when using OPE (Fig. 17). When OPE estimates are not available, the combination of modelling correlated policies as in GP and intelligent policy selection as in UCB gives the best results in each task. Notice the degraded performance of IND+UCB in the first 200 iterations (mostly exploration stage). This happens because each policy is treated independently and until each of them is executed (200 policies) the regret is quite high. Modelling correlation between the policies as in GP methods helps to alleviate this problem.

## C.3 Atari

Similar to Fig. 5, Fig. 19 shows the comparison between our proposed method A-OPS, completely offline policy selection with OPE and completely online selection with IND+UNIFORM on each of 3 Atari games that we consider. Due to the variance of the returns of the policies in this domain, it takes a large number of environment interactions for an online policy evaluation method to provide accurate estimates and in all environments offline evaluation is better. However, A-OPS method outperforms other baselines with only a small amount of environment interactions.

Similar to Fig. 6 (first row), Fig. 20 shows the contribution of each of the components of the method. Our method A-OPS is preferable in all environments across a wide range of interaction budgets. The same observation as before holds: modelling correlated policies as in GP performs better than

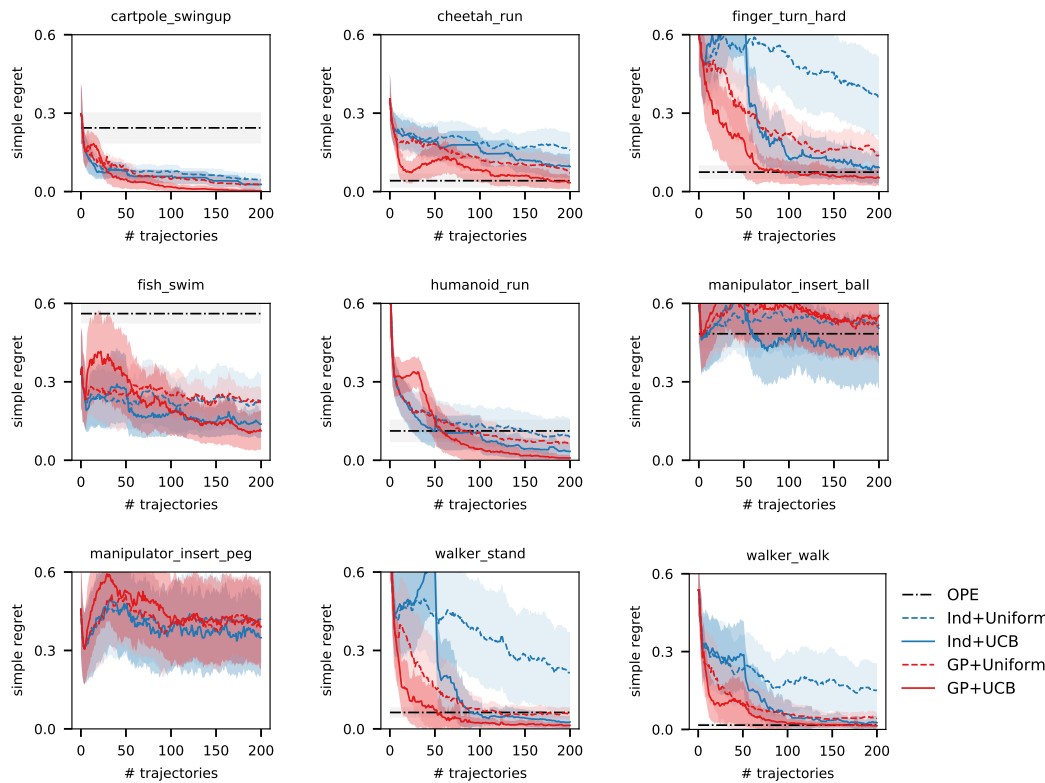

Figure 15: Ablations. In 9 environments of dm-control domain, we ablate two components of our approach when NOT using OPE metrics. 1) The choice of policy model: GP (red) and independent (blue). 2) The policy selection strategy: active (dark, solid line) and uniform sampling (bright, dashed line). We do not show OPE performance in `manipulator_insert_peg` in order to respect the y-axis limits. The value is $0.88$ as D4PG policies are highly overestimated. In general, active is better than uniform, methods that exploit policy correlation are better than methods that do not.

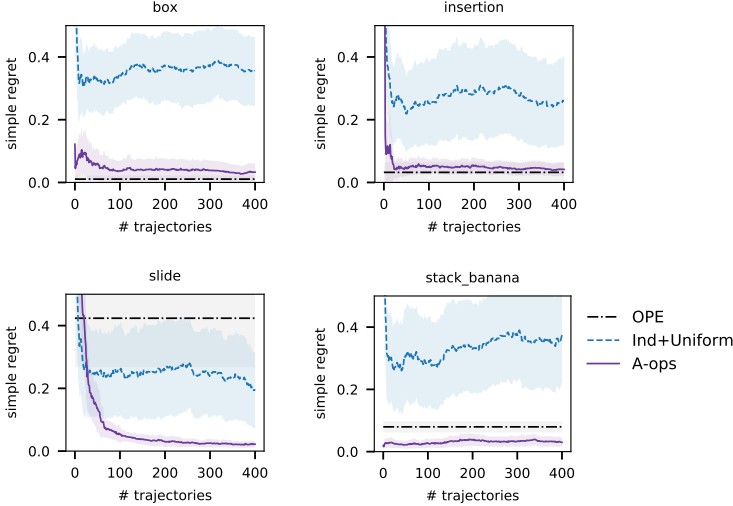

Figure 16: Comparison of A-OPS approach, offline policy selection with OPE, and online policy selection with IND+UNIFORM on each environment in MPG domain. The results are averaged across 100 experiments. Shaded areas show standard deviation of the mean. Our method A-OPS quickly surpasses IND+UNIFORM baseline. A-OPS achieves the regret close to $0$ in all environments.

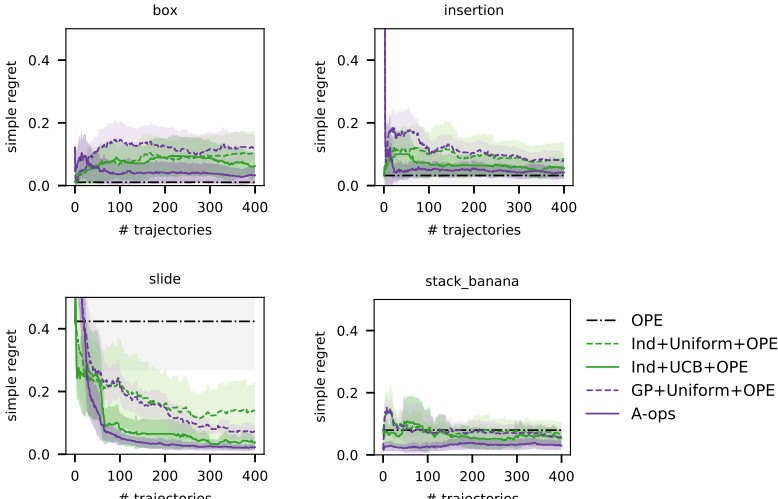

Figure 17: Ablations. In 4 environments of MPG domain, we ablate two components of our approach when using OPE metrics. 1) The choice of policy model: GP (purple) and independent (green). 2) The policy selection strategy: active (dark, solid line) and uniform sampling (bright, dashed line). In general, active is better than uniform, methods that exploit policy correlation are better than methods that do not.

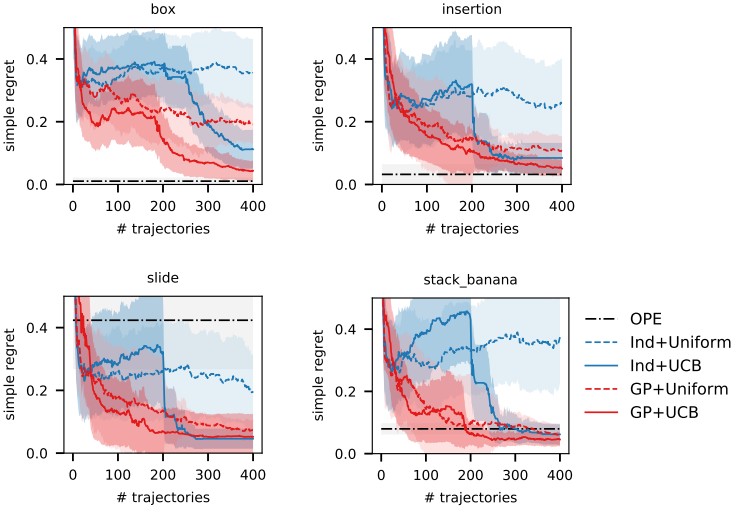

Figure 18: Ablations. In 4 environments of MPG domain, we ablate two components of our approach when NOT using OPE metrics. 1) The choice of policy model: GP (red) and independent (blue). 2) The policy selection strategy: active (dark, solid line) and uniform sampling (bright, dashed line). In general, active is better than uniform, methods that exploit policy correlation are better than methods that do not.

modelling independent policies as in IND, active policy selection as in UCB is better than uniform policy selection as in UNIFORM. Notice that the gain of A-OPS is particularly pronounced in this domain. We attribute it to the data efficiency of our method which is particularly important given a large diversely performing set of policies.

Similar to Fig. 6 (second row), Fig. 21 shows the contribution of each of the components of the method in case when OPE is not used. As before, the results are noticeably worse than when using OPE (Fig. 20). When OPE estimates are not available, the combination of modelling correlated policies as in GP and intelligent policy selection as in UCB gives the best results in each game. Notice the degraded performance of IND+UCB in the first 200 iterations (mostly exploration stage). This happens because each policy is treated independently and until each of them is executed (200

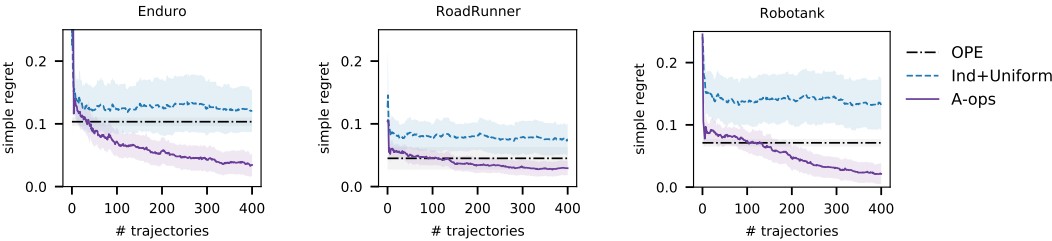

Figure 19: Comparison of A-OPS approach, offline policy selection with OPE, and online policy selection with IND+UNIFORM on each environment in Atari domain. The results are averaged across 100 experiments. Shaded areas show standard deviation of the mean. Our method A-OPS quickly surpasses IND+UNIFORM baseline. A-OPS achieves the regret close to 0 in all environments.

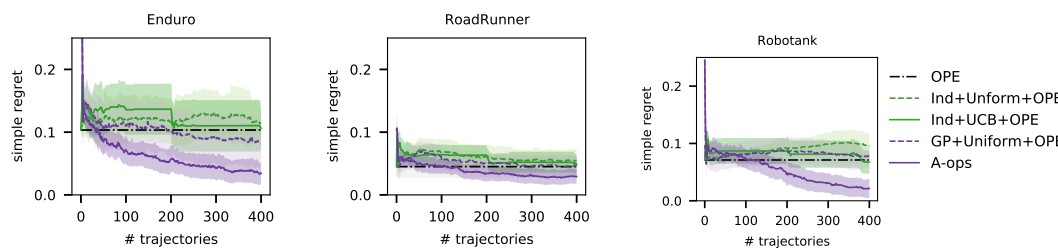

Figure 20: Ablations. In 3 environments of MPG domain, we ablate two components of our approach when using OPE metrics. 1) The choice of policy model: GP (purple) and independent (green). 2) The policy selection strategy: active (dark, solid line) and uniform sampling (bright, dashed line). In general, active is better than uniform, methods that exploit policy correlation are better than methods that do not.

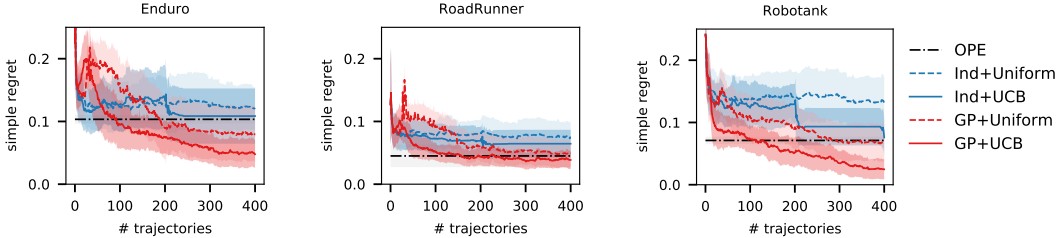

Figure 21: Ablations. In 3 environments of Atari domain, we ablate two components of our approach when NOT using OPE metrics. 1) The choice of policy model: GP (red) and independent (blue). 2) The policy selection strategy: active (dark, solid line) and uniform sampling (bright, dashed line). In general, active is better than uniform, methods that exploit policy correlation are better than methods that do not.

policies) the regret is quite high. Modelling correlation between the policies as in GP methods helps to alleviate this problem and GP+UCB method outperforms all competitors within a small number of trajectories, including OPE method that relies on the offline dataset.

## C.4 Discussion of the results by task

The performance of the methods varies from task to task. We highlight that A-OPS does as well as or better than both offline policy selection (OPE) and online policy selection on 9 of 9 dm-control suite tasks, in 2 of 4 MPG tasks and in 3 out of 3 Atari games. The simple regret of A-OPS approaches 0 or is very low in 7 out of 9 dm-control tasks, in 4 out of 4 MPG tasks and in 3 out of 3 Atari tasks. Thus, in cases when A-OPS regret is not very low, it is still much better than that of the alternative methods, and when A-OPS does not outperform some of the methods, its regret remains very low. As a result, despite variability in different tasks, A-OPS is a very reliable method for policy selection. Another reason for variability across different tasks is the fact that we do not address the question of when to stop active policy selection. In some cases, the OPE estimate is so good that there is

no need for extra online data (for example, as in some of the MPG environments) and in some cases due to the variance of the returns and low correlation of OPE scores with policy returns, even more extensive data collection could be needed. Future work can address this question and derive some recommendations on when to stop policy querying based on statistics in collected data: policy variance, OPE correctness, gap between the policies etc.

In general, the main factors that affect the viability in the A-OPS performance are 1) the distribution of OPE estimates vs ground truth, 2) the variance of returns of a single policy, 3) variance in the returns across available policies. As this information is not available prior to conducting the experiments, it is important that our method A-OPS performs reliably even in the challenging conditions.

Note that the efficiency of A-OPS does not depend on the difficulty of training a policy. For example, training policies from visual input such as in MPG and Atari domains as well as some challenging tasks in dm-control including `humanoid_run`, `manipulator_insert_ball`, `manipulator_insert_peg` require thousands of trajectories to be solved online, but it takes only around 100 of trajectories to identify a good policy.

## D   Qualitative results by task

We illustrate the behaviour of A-OPS by showing its estimates and selection in one experiment in each environment in Fig. 21 in the same way as in Fig. 8. Each experiment is produced with a fixed random seed and the examples are not curated. Thus, they include both the examples of success in identification of the best strategy (for example, among others in `cartpole_swingup`, `insertion`, and `Robotank`) and failure (as in, for example, `manipulator_insert_ball`). For visualisation purposes, we use only 20 policies which are ordered by their ground truth returns (not observed by the method) and perfect predictions would be aligned along a diagonal line. We show the initial *OPE* values, current *selection* of the algorithm, *current* observation, past *observations* and the algorithm *prediction* with standard deviation for steps 1, 10, 50, and 100.

We can see that the initial selection is dominated by OPE prediction. Policy selection by OPE may be quite far from the best policy (for example, as in `manipulator_insert_ball`, `manipulator_insert_peg`, `insertion`, `slide`, `Enduro`), but it may also be very precise leaving no space for further improvement (for example, in `cheetah_run`, `walker_walk`). Notice that A-OPS prediction may be different from observations because our GP model assumes policies are correlated as defined by the policy kernel. An obvious example of this occurs at step $= 1$ when there are no online trajectories and only OPE scores are available. Intuitively, OPE scores are "smoothed" by the kernel when making a prediction. In practice it means that A-OPS predictions at the very few first interactions might be worse than OPE scores if OPE is highly accurate, but they rapidly improve as more online evaluations are collected and OPE is quickly outperformed. If we wanted to match the predictions of OPE at the very first iteration, we could perform the GP optimisation for many iterations and without any hyperparameter priors, but this would essentially result in overfitting and would not have any positive effect on the performance of our method with a moderate amount of online observations.

With more online observations, the prediction by A-OPS improves. We clearly see that the selected policies are not uniformly sampled and are biased toward more promising samples (there are usually more pink points towards the right of the figure). Furthermore, the predictions are interdependent, we can see that the predictions of the return of a particular policy might change even without any new observations of it.

In some experiments the visualisation shows that the predictions by the model are dominated by the variance of the episodic returns at some point during the training. For example, it happens in `cheetah_run`, `fish_swim`, `manipulator` tasks, and `insertion` when at some step during the trajectory acquisition, all predictions look rather "flat" as GP does not assign very different predictions to them given a small amount of observations, but with high variance.

In some experiments we do observe consistent overestimation (for example, in `cheetah_run`, `humanoid_run`, and `walker` tasks) and underestimation (for example, in `Enduro` and `RoadRunner`) by OPE methods. However, A-OPS method still performs well and is able to select a good policy for recommendation. It can recover quickly from this bias thanks to its kernel that brings policies with

similar actions together: When a return is observed for one policy, the predictions for several related policies are adjusted automatically.

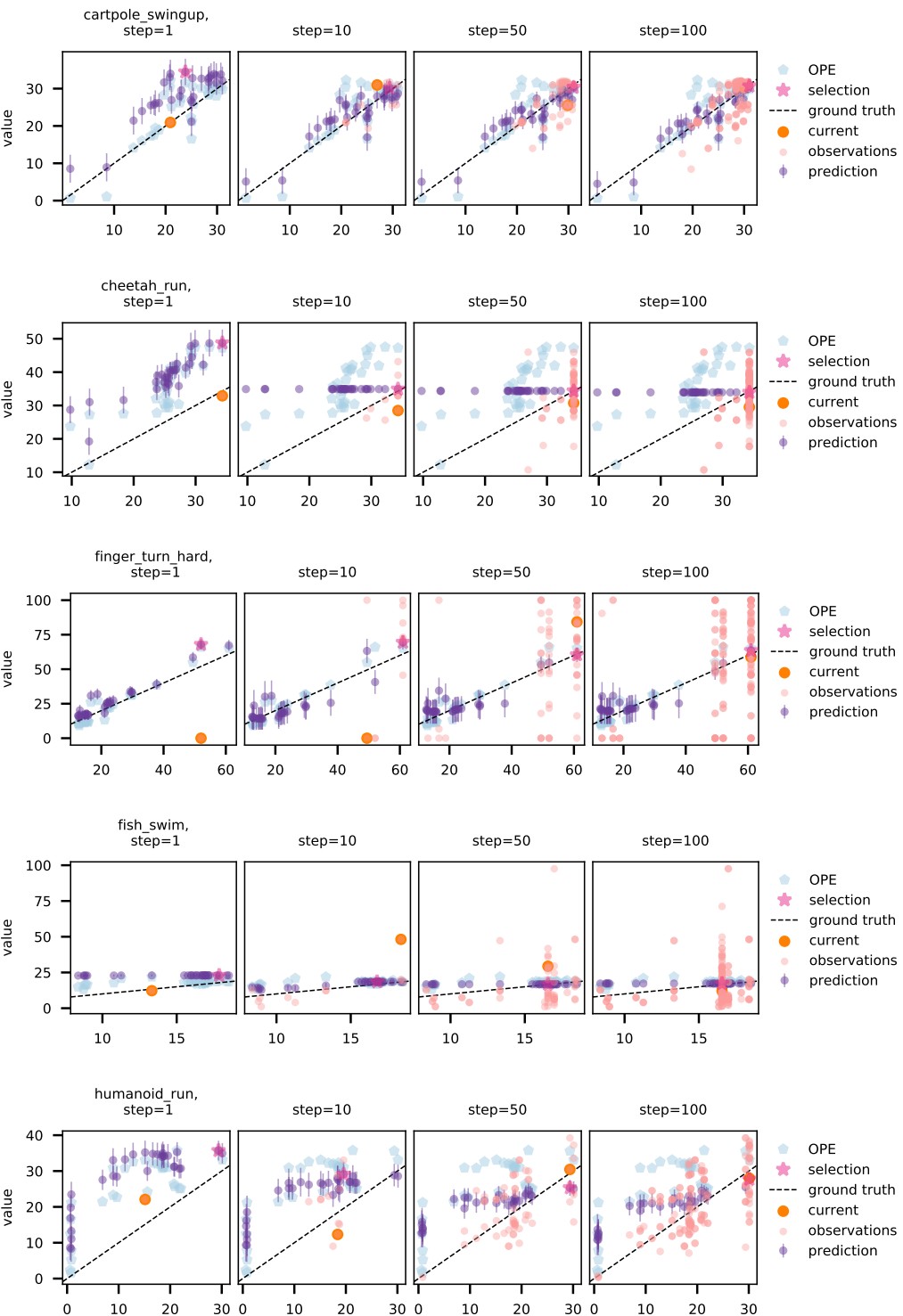

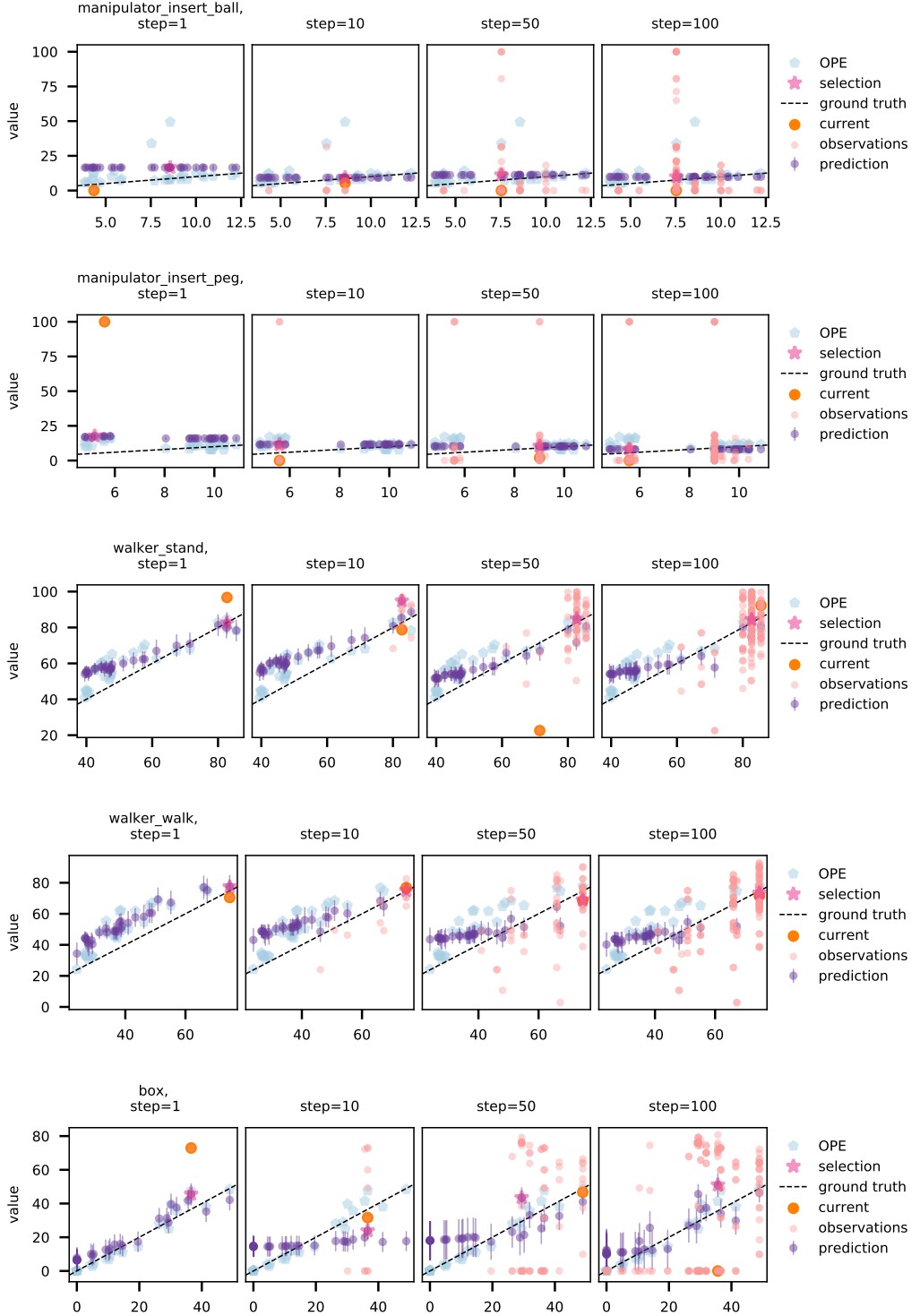

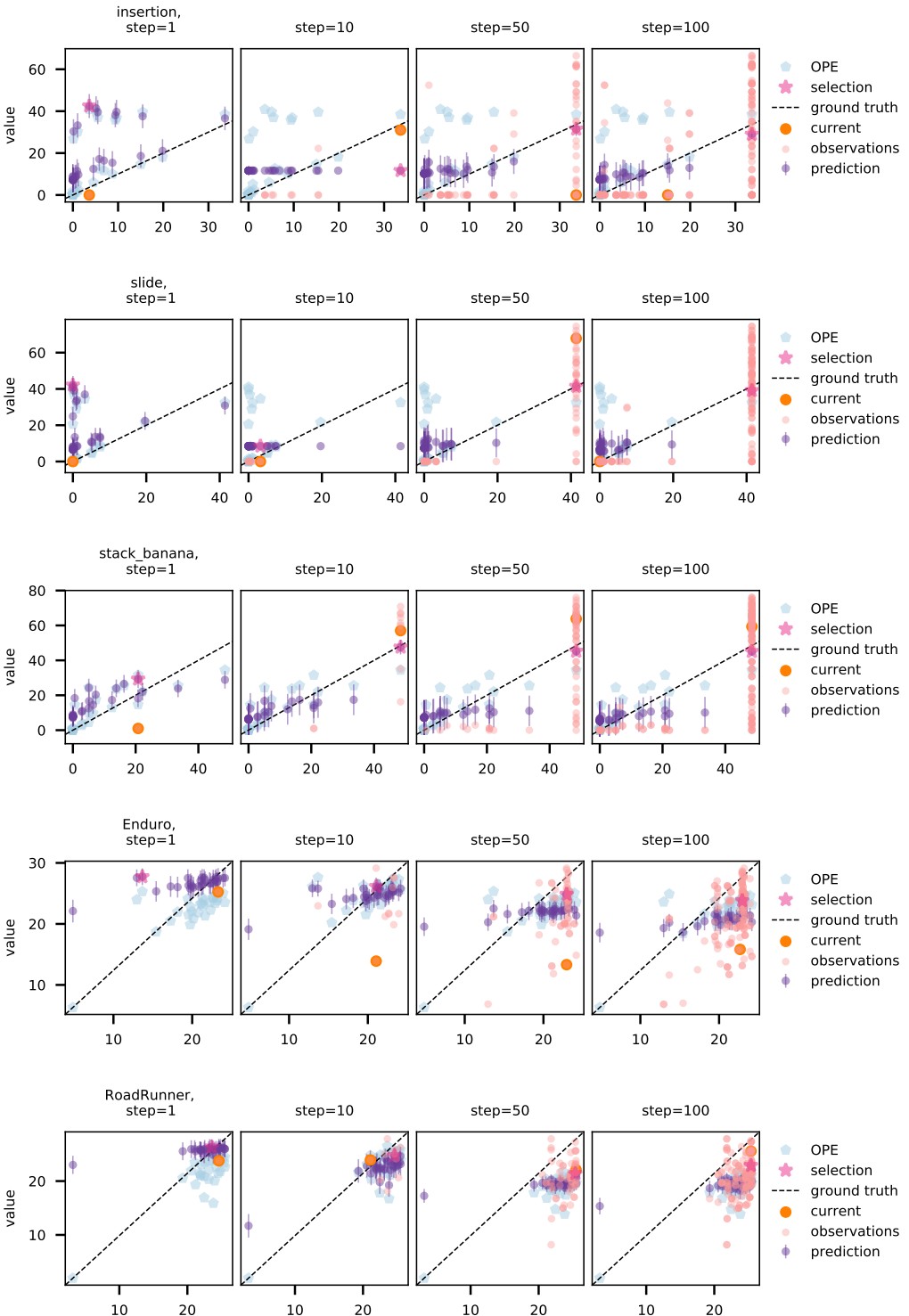

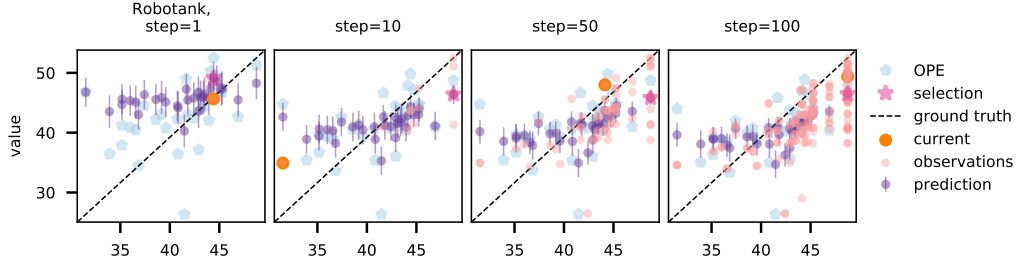

Figure 21: Visualisation of the qualitative performance of A-OPS algorithm on all the environments from dm-control, MPG and Atari domains with 20 randomly selected policies (ordered by ground truth only for the illustration purpose). From left to right we show the state at 1, 10, 50, 100 algorithm iterations. The initial OPE scores are shown in light blue, the policy selected by A-OPS is highlighted with a magenta star, the current observation by the algorithm is shown in orange circle, past observations are in pink and the prediction with its standard deviation is in purple.

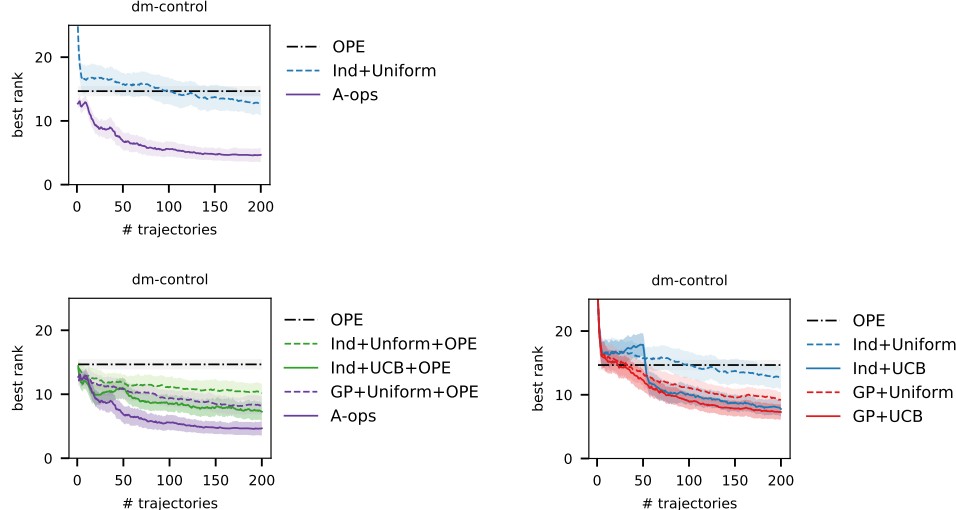

Figure 22: Ranking performance metric. Aggregated across 9 environments of the dm-control domain, we show the performance of the policy selection methods in terms of the rank of the selected policy. The same observations hold: A-OPS is the best strategy, active is better than uniform, methods that exploit policy correlation are better than methods that do not.

## E   Additional experiments

Finally, in this section we present additional experiments that demonstrate various aspects of the problem and our proposed method.

### E.1   Ranking as performance measure

We quantified the performance of the policy selection algorithms based on the simple regret so far. Other performance metrics could be used, for example, the rank of the selected policy. The best rank is 1 that means that the best policy was found and the worst rank is $K$, where $K$ is the number of candidate policies. Fig. 22 shows results similar to Fig. 5 and Fig. 6 for the ranking metric. The same observations still hold.

### E.2   Cumulative and minimum regret

While the goal of A-OPS is to minimise the simple regret and not cumulative regret, it still avoids repetitive execution of most poorly performing policies when employing UCB criterion. As a result,

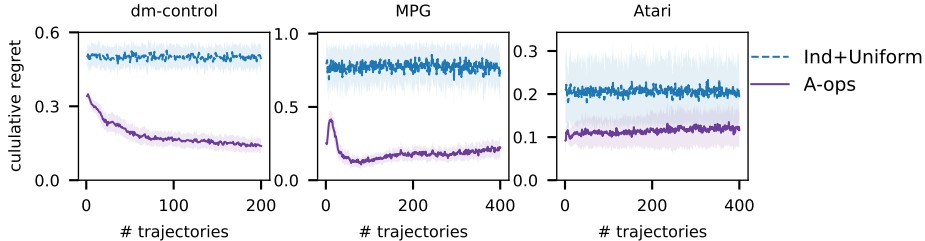

Figure 23: Cumulative regret of A-OPS is lower than that of online policy selection with IND+UNIFORM.

Table 5: The percentage of time that the worst and one of the 10% worst policies is executed during the policy selection procedure by different algorithms including A-OPS.

| method | dm-control | MPG | Atari |
|---|---|---|---|
| | the worst policy | | |
| IND+UNIFORM+OPE | 2% | 0.52% | 0.52% |
| IND+UCB+OPE | 1.2% | 0.26% | 0.31% |
| A-OPS | 0.14% | 0.007% | 0.01% |
| | 10% worst policies | | |
| IND+UNIFORM+OPE | 12% | 10.6% | 10.5% |
| IND+UCB+OPE | 7.7% | 5.4% | 9% |
| A-OPS | 1.3% | 0.17% | 1.4% |

A-OPS achieves lower cumulative regret than strategies based on uniform sampling as manifested by Fig. 23. Interestingly, towards the end of the experiment the cumulative regret starts growing slightly, that indicates that A-OPS goes back to the exploration phase again once a promising policy is identified.

We also analyse how often during the policy selection procedure A-OPS executes very bad policies. For this we compute the percentage of A-OPS (and alternative methods) iterations where the worst policy or one of the 10% worst policies is executed. Our findings are summarized in Tab. 5 for the three most representative algorithms. A-OPS strategy ends up querying the worst-performing policies significantly less frequently than other algorithms despite not being optimized for it directly. A potential explanation is that the policy kernel allows us to infer which policies are poorly performing in a very data efficient manner.

### E.3 Varying amount of OPE estimated

Computing OPE estimates (like FQE or model-based OPE) can sometimes be expensive. Fig. 24 depicts the performance of A-OPS by providing OPE estimates for a random subset of $k \in [0, 50]$ policies in dm-control environment. Increasing the number of OPE observations consistently improves the results. This result demonstrates that our method A-OPS can seamlessly incorporate all the available information and benefit from the increasing number of initial observations.

### E.4 Effect of the selections strategies

In the next experiment we study the effect of the choice of Bayesian Optimisation method on the performance of the A-OPS method. First, we investigate the influence of the exploration coefficient on the performance. It turns out that the performance of the A-OPS method is not very sensitive to the choice of exploration coefficient value. Our ablation study with various values in Fig. 25, right shows that there is no difference in performance for the values between 1 and 7 and the performance slightly degrades, but still stays better than for the baselines, for the values 7-20.

Furthermore, we note that the selected best strategy A-OPS as GP+UCB+OPE is just one realisation of the active policy selection strategy. We chose UCB as our strategy as it demonstrated promising results across different environments and domains. However, other active selection strategies could

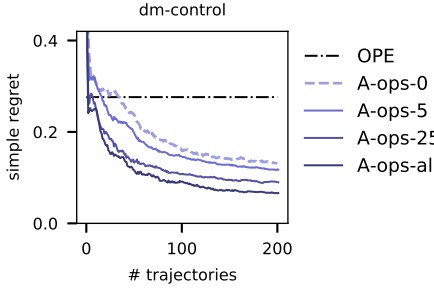

Figure 24: The performance of A-OPS on dm-control with varying number of FQE

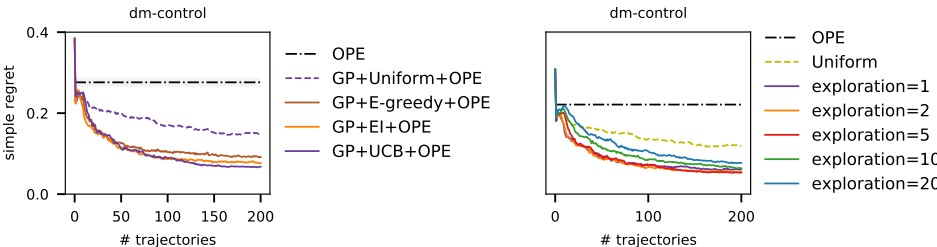

Figure 25: Left: The performance of A-OPS where different values of exploration coefficients are used. The difference between different coefficients is small for a wide range of values and using UCB criterion with any exploration value is better than doing uniform sampling. Right: The performance of various selection strategies in conjunction with GP+OPE in 9 environments of dm-control domain. The difference between uniform and active selection strategies is significant, while the difference between various active selection strategies is small. For better visualization, we do not show the standard deviation, which is comparable to the A-OPS result in Fig. 6.

be applied as well, for example, epsilon-greedy or expected improvement (EI). In Fig. 25, left we show the performance of various selection strategies in one domain. The choice of a BO strategy makes a considerably smaller difference than the increase of regret when using uniform sampling (GP+UNIFORM+OPE).