# OpenReview forum: "Active Offline Policy Selection"
_NeurIPS.cc/2021/Conference — NeurIPS 2021 Poster_

### Official Review · Reviewer_bUXf · 2021-07-14

**Rating:** 7
**Confidence:** 4

**Summary:**

This paper considers a novel problem setting of "Offline Policy Selection", whereby there is access to an initial set of policies trained offline, and an algorithm must choose a policy to deploy in the online simulated environment. I am giving this paper the score of weak reject because despite introducing an important problem setting, I am not convinced the method is actually particularly effective. Hopefully this can be clarified during the rebuttal phase. As such, the emphasis of this review is on areas for improvement.

**Ethics Review Area:**

["I don’t know"]

**Limitations And Societal Impact:**

Limitations are discussed sufficiently well but of course could always be expanded. There is no expected negative societal impact from this work.

**Main Review:**

Strengths:

1) Novel problem setting, definitely well motivated. Could be important in using Offline RL more broadly.

2) The policy kernel idea seems intuitive, and the discussion around choice of kernels (A.5) is interesting. I think it would improve the paper if some of this discussion/additional experiments are included in the main body (more below).

3) The authors did make efforts to include multiple ablations, and discussed limitations.

4) The use of OPE + online is obvious in hindsight which is a common trait of valuable contributions.

Weaknesses/Questions:

1) The actual number of trials used is much larger than I would expect, with BO not converging until >100 trajectories. For DM control suite, *this is enough to train a near optimal policy from scratch using online data*. Why would we not just do this? I think a key baseline is OPE+online, since OPE always performs reasonably well zero-shot, we could then use the online data to fine tune the best OPE policy.

2) Given 1), it seems that formulating this problem as blackbox optimization may not be the best way to go, since it is not making use of the structure of the task at all. Aside from using the episode returns, A-OPS does not leverage any of the trajectory data (as far as I am aware) which ends up being 200k timesteps worth of online samples by the end.

3) What is the actual variance of the policy returns? The aggregate regret is definitely an effective overall metric to use, but if for example the min and max polices for a task are only 5-10% apart it is rather meaningless.

4) It would be useful to include a plot showing the *min return* for each method. Using an exploratory acquisition function (such as UCB) likely leads to also testing some policies far away from the current best in the behavioral space, which could lead to very poor returns. For practical use this may be infeasible since if deploying on real hardware you could break the robot. This is alluded to in the Limitations section in the Conclusion, but it is not shown anywhere. I have a minor concern this may be because it looks bad for A-OPS.

5) Looking at the *individual* task performance in the Appendix, it seems to be very hit and miss. Do the authors have any intuition why none of the methods or ablations work well for the manipulator dm control task? Similarly with MPG, for 2/4 of the tasks the OPE approach actually does better than A-OPS despite using zero online data. It seems only really the slide task which makes A-OPS outperform overall due to the poor performance of OPE, what happened there?

6) How did you tune the UCB exploration parameter?

7) I assume the answer to 6) and also other ablations is that the method was tuned on the testing data. Is this true?

Minor comments/typos:

-l.97: "We can use GP" → "we can use a GP"

-l.204: "We perform 4 several manipulation tasks" (not clear what this means)

**Time Spent Reviewing:**

6

---

> ### Author Response · Authors · 2021-08-10
> **Reply to reviewer bUXf**
>
> We would like to thank you for the thoughtful review that will help us to revise the paper. We will address common questions in the general answers. Below we address the other concerns.
>
> 1. *"The actual number of trials used is much larger than I would expect, with BO not converging until >100 trajectories. For DM control suite, this is enough to train a near optimal policy from scratch using online data."*
>
> It is a good point and let us clarify it. While some DM control suite tasks can be solved in a relatively small number of episodes, other tasks cannot, for example high-dimensional control in Humanoid Run, Manipulator Insert Ball, and Manipulator Insert Peg require 100k or more episodes to learn. Additionally we consider the MPG domain with robotic manipulation policies trained **from pixels** which also requires significantly more than 100 episodes (8000 with additional human demonstrations [66]). These examples highlight that the efficiency of a-ops does not depend on the difficulty of training a policy. To verify this, we ran additional experiments and applied a-ops to select vision-based policies in Atari games that required thousands or more trajectories to train a good policy. We can still achieve low regret within 100 episodes. We will include it in the appendix if the reviewer considers it useful.
>
> Additionally, even data efficient RL algorithms require hyperparameters to be set properly. In the DM control suite, the prior work can guide us to choose a good hyperparameter set. However, for new domains online training may have to run multiple times to find good hyperparamteters.
>
> *"I think a key baseline is OPE+online, since OPE always performs reasonably well zero-shot, we could then use the online data to fine tune the best OPE policy."*
>
> Starting from the policy selected by OPE and fine-tuning it will not address the problem of policy selection, but it is an interesting direction for future research. One challenge is that the policy selected by OPE metric could be very bad. For example, in 3 out of 9 environments of DM control it is closer to the performance of the worst policy than to the best policy. Furthermore, OPE+fine tuning is not a straightforward baseline as many questions need to be resolved, such as: what parameters to fine tune? for how long? how to weigh past and new data? how to collect new data: from the best policy, uniformly or intelligently? etc.
>
> 2. *"formulating this problem as blackbox optimization may not be the best way to go, since it is not making use of the structure of the task at all. "*
>
> We would like to note that a-ops already uses some structure of the task such as, for example, it considers two types of observations (OPE and policy returns) in the GP and a special form of the kernel for policy comparison. The rest of the question is covered in general answers.
>
> 3. *"What is the actual variance of the policy returns? The aggregate regret is definitely an effective overall metric to use, but if for example the min and max polices"*
>
> In most datasets there is a significant gap between the scores of the bad performing and well performing policies. Here are the standard deviation, min and max returns of the policies from our dataset:
>
> | task                    | stdev  | min return | max return |
> |-------------------------|--------|------------|------------|
> | cartpole_swingup        | 6.488  | 0.451      | 31.504     |
> | cheetah_run             | 5.504  | 6.933      | 36.291     |
> | finger_turn_hard        | 14.055 | 8.613      | 60.971     |
> | fish_swim               | 3.208  | 8.097      | 18.755     |
> | humanoid_run            | 8.993  | 0.566      | 30.117     |
> | manipulator_insert_ball | 2.825  | 3.668      | 13.801     |
> | manipulator_insert_peg  | 2.306  | 4.668      | 11.249     |
> | walker_stand            | 13.173 | 39.180     | 85.533     |
> | walker_walk             | 13.973 | 11.950     | 74.719     |
> | box                     | 19.711 | 0.0        | 64.58      |
> | insertion               | 15.808 | 0.0        | 52.42      |
> | slide                   | 14.622 | 0.0        | 52.17      |
> | stack_banana            | 18.420 | 0.0        | 58.74      |
>
> The gap between the policies and their variance are significant, thus, the policy selection by a-ops can bring a significant benefit.
>
> 4. *"It would be useful to include a plot showing the min return for each method."*
>
> The minimum return of the deployed policy is linked to the cumulative regret. While the goal of a-ops is to minimise the simple regret and not cumulative regret, it still avoids repetitive execution of most poorly performing policies when employing UCB criterion. As a result, a-ops still achieves lower cumulative regret than strategies based on uniform sampling. If the cumulative reward is the main concern in some applications, we could adapt our method and choose a BO/bandit algorithm with this aim. More on safety considerations in the general answers.
>
> 5, 6. *"How did you tune the UCB exploration parameter?"*  *"the method was tuned on the testing data. Is this true?"*
>
> No, our method wasn't tuned on the testing data. We set the GP hyperparameters based on the statistics about the policy returns from the offline dataset. For the UCB exploration parameter, we experimented with the values based on the cartpol_swingup environment, then fixed it for all the other tasks and domains. In fact, regret is not very sensitive to the choice of exploration coefficient. To verify it, we ran an ablation study with various values. There is no difference in performance for the values between 1 and 7 and the performance slightly degrades, but still stays better than for the baselines, for the values 7-20.
> Moreover, our method is not specific to the UCB method. As we have shown in Figure 17, other BO methods perform comparatively and for example, EI (Expected Improvement) does not require any parameters.

---

> > ### Comment · Reviewer_bUXf · 2021-08-13
> > **Increasing my score**
> >
> > I have read the response and think it is now possible to increase my score to a 6, on the basis that the content we discussed would be added to the Appendix. My two remaining comments are:
> >
> > 1. Cumulative regret is not a perfect proxy for min return. I am curious to know if during the process of selecting the best policy, your method actually selects the *worst* policy more frequently than other methods. I know you are not optimizing with safety constraints, so it is fine if your method is the worst here, but I figured it would be important to show as a limitation/area for future work. In general showing more information about the method only improves the paper in my opinion.
> >
> > 2. Are there plans to open source the code? From my perspective, the most valuable contribution of this paper is the new problem setting, so to make it possible for the community to build on the results the code would be very important.
> >
> > The increase in score is not contingent on positive responses to these questions, and in addition I will pay attention to other discussions to consider further increasing if it is possible/required.
> >
> > Thank you!

---

> > > ### Author Response · Authors · 2021-08-23
> > > **Thank you and response to comments**
> > >
> > > Thank you for your feedback and for increasing the score. Here are our answers to the remaining questions.
> > >
> > > 1. Indeed cumulative regret is not a perfect replacement for min return. To understand the minimum return in the experiments, we now computed the percentage of a-ops (and alternative methods) iterations where the worst policy or one of the 10% worst policies was queried. Our findings are summarized in the following table for the three most representative algorithms. A-ops strategy ends up querying the worst-performing policies significantly less frequently than other algorithms despite not being optimized for it directly. A potential explanation is that the policy kernel allows us to infer which policies are poorly performing more data efficiently.
> > >
> > >
> > > | domain             	| dm-control 	| MPG    	| Atari 	|
> > > |--------------------	|------------	|--------	|-------	|
> > > |  **the worst policy**  	|            	|        	|       	|
> > > | Ind+Uniform+OPE    	| 2%         	| 0.52%  	| 0.52% 	|
> > > | Ind+UCB+OPE        	| 1.2%       	| 0.26%  	| 0.31% 	|
> > > | A-ops                     	| 0.15%      	| 0.007% 	| 0.01% 	|
> > > | **10% worse policies** 	|            	|        	|       	|
> > > | Ind+Uniform+OPE    	| 12%        	| 10.6%  	| 10.5% 	|
> > > | Ind+UCB+OPE        	| 7.7%       	| 5.4%   	| 9%    	|
> > > | A-ops                     	| 1.3%       	| 0.17%  	| 1.4%  	|
> > >
> > >
> > > 2. We agree that open-sourcing active OPS code would be very valuable, and we will try to open-source the core components of our algorithm for the camera-ready.

---

> > > > ### Comment · Reviewer_bUXf · 2021-08-25
> > > > **Appreciate the response**
> > > >
> > > > Hi,
> > > >
> > > > Thank you for this. I already increased to a 6 and would have been happy to see the paper accepted. However, if you can include this result in the Appendix, I am happy to increase to a 7.
> > > >
> > > > The point is not so much that it is a good result, which is good to know of course, but more so that I felt it was missing from the discussion.
> > > >
> > > > It would also be fantastic if you could open source as much as possible, of course within the usual constraints.
> > > >
> > > > I don't see much potential to increase further, but I definitely think this is now a solid contribution as all of the grey areas have been cleared up from my perspective.

---

> > > > > ### Author Response · Authors · 2021-08-26
> > > > > **Thank you and we are happy to include new results**
> > > > >
> > > > > Dear reviewer, thank you!
> > > > >
> > > > > Indeed this is a valuable insight into the method and we appreciate your thoughtful feedback. We have now prepared an updated manuscript that includes this discussion (as well as other important points raised in the reviews) and we will post it when possible.

---

### Official Review · Reviewer_rn9y · 2021-07-16

**Rating:** 5
**Confidence:** 5

**Summary:**

This paper works on the policy selection problem in a setting characterized by features: available logged data and allowing a limited number of online interactions with the environment. To do so, this work proposes to use Bayesian optimization to combine existing OPE techniques and online interactions. For the setting with large number of policies, a kernel function is introduced to take the correlation between policies, which indicate the performances of different policies.

**Limitations And Societal Impact:**

Please also refer to the main review section. For high-stakes decision-making scenarios that offline RL aims to address, further online interactions with the environment is not available for reasons including ethical or cost concerns. Take healthcare and education for example, logged dataset can be large, but online interaction is typically infeasible. This work fails to show the importance of the proposed setting itself.

**Main Review:**

The problem formulation is not typical. It is a mix of the common offline and online settings. On the one hand, it aims to leverage the logged datasets and OPE; on the other, it is not totally offline as the setting in the paper allows online interactions, although the number such interactions are limited. Given this, this work proposes a Bayesian optimization solution. While such a setting is possible in real-world, I doubt the importance of such a non-typically setting itself.

**Time Spent Reviewing:**

2.5

---

> ### Author Response · Authors · 2021-08-10
> **Reply to reviewer rn9y**
>
> Thank you for taking time to review our paper.
>
> In response to your main concern: *"a mix of the common offline and online settings" "I doubt the importance of such a non-typically setting""further online interactions with the environment is not available for reasons including ethical or cost concerns" "This work fails to show the importance of the proposed setting itself."*
>
> We agree that this is a new problem setting that has not yet been studied in the literature. It is true that a-ops (and deep RL more generally) may not be applied to certain safety critical domains, where even a small budget of interactions is infeasible.
> However, we maintain there are important real-world applications that do allow a modest interaction budget. In these situations a-ops can help to reduce the amount of required interactions. These applications include robotics and A/B testing.
>
> Furthermore, the other reviewers find the motivation to be a strong point. For example, 68yo: *"Indeed much of the recent focus on offline RL and off-policy evaluation has neglected the possibility for a small budget of interaction with the environment."*, 6fix: *"well-motivated by practical situations such as A/B tests"*, bUXf: *"Novel problem setting, definitely well motivated. Could be important in using Offline RL more broadly."*
>
> We hope that we can contribute to the community by studying a novel problem setting that may be useful for **some** important real-world domains. In this setting we propose a novel method that outperforms existing methods.

---

> > ### Comment · Reviewer_rn9y · 2021-09-01
> > **Thanks for the resonses and follow-up**
> >
> > I would like to thank the authors for addressing my concerns. I have two follow-ups based on the responses. First of all, there is no problem at all for proposing a new setting. The real concern is NOT whether this setting is new but the underlying motivations. While it is true that there are real-world cases (I totally agree with this) where a few online interactions are possible, it is straightforward to adjust a good (hopefully effective) totally offline RL algorithms to such scenarios by conducting a limited number online interactions. In this light, it is crucial for the community to focus on developing powerful offline RL algorithms from logged dataset, which can be implemented in the mixed setting. Secondly, I do read comments by other reviewers very carefully. However, I think reviewers are allowed to have different views. Thanks again for highlighting those comments, but maybe that should not be the reason for me to change (otherwise, we only need one reviewer for each paper).

---

### Official Review · Reviewer_6fix · 2021-07-17

**Rating:** 7
**Confidence:** 4

**Summary:**

This paper introduces a problem of active offline policy selection, where the goal is to select the best-performing policy among the fixed K policies using an abundant offline dataset and a limited additional interaction with the environment. This work proposes to solve this problem by Bayesian optimization (BO) with a Gaussian process (GP), where the input and output of the objective function are a policy and its expected return respectively, and function queries render noisy observations. To facilitate GP in this problem, a new GP kernel is introduced, which takes two policies as input, and its output is computed based on the similarities of policy distribution for each state in the offline dataset. Also, to make BO more data-efficient, off-policy evaluation (OPE) estimates through the offline dataset are used for a warm start of BO. Finally, UCB acquisition function is used for active policy selection. Experimental results show that the proposed BO with OPE-based warm start significantly outperforms the baselines. Also, ablation studies confirm that UCB selection is better than uniform selection, GP is better than methods that do not consider policy correlation, and using OPE is better than not using it.


**Ethical Concerns:**

-

**Limitations And Societal Impact:**

I think the authors have discussed the limitations and potential negative societal impact properly.

**Main Review:**

- Overall, the paper is well-written and easy to follow. The introduced active offline policy selection is well-motivated by practical situations such as A/B tests, and the proposed method is sound with a suitable combination of Bayesian optimization and OPE estimates. I found introducing this setup is novel. The proposed solution itself is not surprising, but the experiments are thorough and the results are also convincing. The ablation studies are solid and appreciated.
- The additional data collected during online interactions could be used to improve the OPE estimates since they are particularly on-policy samples for a candidate policy?
- In Eq (3), the generative assumption may not be consistent with the actual episodic return distribution since it is likely that return distribution may not be Gaussian-distributed (e.g. highly skewed distribution). Would this inconsistency make a problem?
- In Figure 3 (middle), why are the similarities between D4PG policies very low, even with the same hyperparameters?
- It seems the proposed policy distance (Eq (6)) may not be optimal since it considers state distribution by the offline dataset, not the state distribution induced by the target policy. For example, if the target policies visit a very limited region of states, to measure the policy similarity, we only need to consider those states that will be visited while ignoring other states that will not be visited. The distance between stationary distributions induced by policies could be a more suitable distance measure?
- What if the OPE estimates are significantly underestimated (or overestimated), compared to the true expected returns? Pure exploitation (or pure exploration) might happen during active policy selection?
- In Figure 5, why doesn't the performance of A-opt at (# trajectories=0) match the performance of OPE (even worse than OPE)?


**Time Spent Reviewing:**

4

---

> ### Author Response · Authors · 2021-08-10
> **Reply to reviewer 6fix**
>
> Thank you for the detailed review. We will revise the manuscript to clarify the raised questions. We answer the repeating questions in the general answers and here we address the rest.
>
> *Assumption about the Gaussian distribution of the returns*
>
> It is common to assume a Gaussian noise distribution in Bayesian optimization when the function value is a real number. It is true that in most environments the distribution of episodic returns is non Gaussian (could be bimodal, skewed etc.) However, empirical results show that this inconsistency does not cause any particular problem. Nonetheless, if the return noise distribution is known in advance, it can replace the Gaussian noise model and approximate GP inference can be performed.
>
> *"In Figure 3 (middle), why are the similarities between D4PG policies very low, even with the same hyperparameters?"*
>
> Indeed we observe that in many cases D4PG policies are quite different from each other. A few related observations support this finding.
> - It has been shown[71] that the performance of a related algorithm (DDPG) in online training is quite sensitive to random seed initialisation. Thus, it is not surprising that our kernel indicates that the agents' actions are quite different from each other.
>
>     [71] Deep Reinforcement Learning that Matters. Peter Henderson, Riashat Islam,Philip Bachman, Joelle Pineau, Doina Precup, David Meger. AAAI-18.
> - The other policies in Figure 3 (middle) use regularization towards the behavior prior. D4PG does not and the trained policies can be quite far from the behavioral policy as shown in Figure 3 (right). Thus, D4PG policies have more space to be different from each other than the regularized policies that stay close to the behavioral policy.
>
>
> *"The distance between stationary distributions induced by policies could be a more suitable distance measure?"*
>
> In this work we prioritized the simplicity of the kernel computation and even such an approach has shown promising results. It could be a good idea to select the states for computing the kernel intelligently. Methods from the DICE family could be used here, but their accuracy degrades when the evaluation policy is far from the behavior policy. Furthermore, estimating the stationary distribution could be a challenging task, especially with high-dimensional states.
>
> *"What if the OPE estimates are significantly underestimated (or overestimated), compared to the true expected returns?"*
>
> In some experiments we do observe consistent under and overestimation by OPE methods. We will include the relevant illustrations to demonstrate this. However, a-ops still performs well. We believe that a-ops can recover quickly from this bias thanks to its kernel that brings policies with similar actions together: When a return is observed for one policy, the predictions for several related policies are adjusted automatically. In addition, if we have some prior knowledge about consistent under or overestimation, the probabilistic model can be easily extended to include a constant bias parameter to estimate together with other GP hyper-parameters.
>
> *"In Figure 5, why doesn't the performance of A-opt at (# trajectories=0) match the performance of OPE (even worse than OPE)?"*
>
> A-ops prediction may be different from observations because our GP model assumes policies are correlated as defined by the policy kernel. Intuitively, OPE scores are "smoothed" by the kernel when making a prediction. In practice it means that a-ops predictions at the very few first interactions might be worse than OPE scores if OPE is highly accurate, but they rapidly improve as more interactions are collected. As our goal is to select the policy to deploy after several trajectories, we do not attempt to match the performance of OPE at #trajectories=0.

---

### Official Review · Reviewer_68yo · 2021-07-17

**Rating:** 7
**Confidence:** 4

**Summary:**

This paper introduces the problem of active offline policy selection which consists of selecting the best policy from a set of candidates given a small budget to interact with the environment. The paper proposes a solution (A-OPS) based on Bayesian optimization using a kernel function to relate candidate policies and generalize between them. The proposed solution is thoroughly evaluated on simulated robotic control environments using benchmark datasets from the off-policy evaluation literature and several ablations are included to rationalize the choices of the proposed method.

**Limitations And Societal Impact:**

Space is already a bit tight, but the paper would benefit from a more substantial discussion of the limitations in the current methods and the potential directions of improvement. I do not think a discussion of societal impact is necessary for this paper.

**Main Review:**

### Strengths:

1. The proposed problem seems important and not yet formalized in the community. Indeed much of the recent focus on offline RL and off-policy evaluation has neglected the possibility for a small budget of interaction with the environment. Thinking about how to effectively exploit this small budget is an important area for future work, and hopefully this paper can spur research in this area.
2. The proposed solution is elegant. While not ground-breaking, the proposed solution does seem to be an intuitive and elegant way to approach the problem. With such a small budget of interaction, bayesian optimization is often a very strong approach, as this paper bears out.
3. The experimental evidence is substantial. The paper runs on many environments with large datasets of policies. The advantage over the baselines is very evident. The ablations seem to do a comprehensive job of rationalizing each of the major components of the proposed approach.

---

### Weaknesses:

1. The formalism is often imprecise. Here are a few examples:

   (a) In Section 2.1, the notion of a budget which is often discussed informally in the text is never formally defined

   (b) As a result $ \hat k$ is used in the main definition of regret but is never defined. This makes it somewhat unclear if the paper is concerned only with the regret of the final policy or if it also cares about the cumulative regret during the evaluation in the environment (my impression was the former, but slightly unclear).

   (c) In section 2.2 $ N_k $ is never defined, but often used. (It seems to be used to indicate the number of return observations of policy $ k$)

2. Some important details are omitted in the experiments section.

   (a) In Figure 7 the labels on the x-axis are never explained. The reader can try to guess (is MB-AR maybe model-based auto-regressive?), but these labels really ought to be explained in the text.

   (b) For the plots in e.g. Figure 5, are the algorithms re-run from the start given each budget or are the algorithms just progressively evaluated after each trajectory is added? It seems to be the second, but this was a bit unclear.

3. This is perhaps a more minor point, but based on the results in the appendix, it would seem useful to discuss the variability of the performance across tasks. While in aggregate the advantage of A-OPS over simple OPE is obvious, the advantage is much larger on some tasks than others (Fig 9, Fig 12). Moreover, for some tasks in the MPG domain, it seems that A-OPS performs worse than OPE even after 400 trajectories (Fig 12). Even if this discussion is just put in the appendix, it would be useful to discuss rather than just repeating that A-OPS is outperforming the other methods.

---

### Recommendation:

Accept.

I think this is a strong paper that introduces an interesting problem and provides a reasonable solution with substantial experimental evidence. The paper is sloppy about details in a few places, but these should be easily fixed.

---

### Minor comments/questions:

1. Is there a way for the algorithm to incorporate knowledge of the size of the budget? Now the budget just determines the length of the loop, but it seems that the algorithm could modify its strategy with knowledge of the budget.

2. Have you thought about how to select the set of candidate policies to best leverage the strengths of A-OPS? I understand that for this paper





**Time Spent Reviewing:**

4

---

> ### Author Response · Authors · 2021-08-10
> **Reply to Reviewer 68yo**
>
> Thank you for the review and useful suggestions. We answer the repeating questions in the general answers and address the rest of the concerns here.
>
> 1. Thank you for pointing out some inconsistencies with the formalism of the paper. We will further clarify it in the paper.
>     (a) *"the notion of a budget"*: Budget is measured by the number of episodes in our paper.
>
>     (b) *"k^ in the main definition of regret"*: $\hat{k}$ is defined in line 89 as the policy with the highest posterior mean. In our problem formulation, we are mainly concerned about the regret of the final recommended policy.
>
>     (c) *"section 2.2 Nk"*: $N_k$ is indeed the number of return observations of policy $k$, first referred to in line 115.
>
> 2. Thanks for identifying missing details in the experiments section, we will include them in a revision.
>
>     (a) *"Figure 7... labels"*
>
>     Good point, we will add this to the text. For the reference they are:
>     - MB-AR: Model-Based OPE with an auto-regressive transition model
>     - FQE-D: Fitted Q Evaluation with distributional q function and distributional loss
>     - MB-FF: Model-Based OPE with a feedforward transition model
>     - FQE-L2: Fitted Q Evaluation with standard q-function and L2 loss
>     - DR: Double Robust OPE
>     - VPM: Variational Power Method
>     - IS: Importance Sampling
>     - DICE: Distribution Correction Estimation
>
>     (b) *"For the plots in e.g. Figure 5, are the algorithms re-run from the start given each budget or are the algorithms just progressively evaluated after each trajectory is added?"*
>
>     The algorithms are progressively evaluated when each trajectory is added. We will clarify it.
>
> Minor questions
>
> 1. *"Is there a way for the algorithm to incorporate knowledge of the size of the budget?"*
>
>     Yes, it could be possible to incorporate the budget constraints in BO. While it is out of scope of the current paper, a-ops problem could be modified as the best arm identification problem with a fixed budget [4, 24]. In this case, BayesGap method in [24] could be applied directly instead of the UCB.
>
> 2. *"Have you thought about how to select the set of candidate policies to best leverage the strengths of A-OPS?"*
>
>     In this work we assumed that a set of policies is given to us in advance. Possibly, it could be curated by an expert or other algorithm to filter out inappropriate policies, for example, if they are unsafe. No specific curation mechanism is needed for a-ops to operate, it can deal with a large set of policies and quickly eliminate many of them based on their similarities.

---

### Author Response · Authors · 2021-08-10
**General answers**

Here we address the questions that were raised by several reviewers. We answer the particular questions of each reviewer separately.

Reviewers **68yo, bUXf: Variability of the results for separate tasks.**

68yo: *"it would seem useful to discuss the variability of the performance across tasks ..."*

bUXf: *"Looking at the individual task performance in the Appendix, it seems to be very hit and miss ..."*

The performance of the methods vary from task to task. We would like to highlight that a-ops does as well as or better than both offline policy selection (OPE) and online policy selection on 9 of 9 DM Control Suite tasks and in 2 of 4 MPG tasks. The simple regret of a-ops approaches 0 or is very low in 7 out of 9 DM Control Suite tasks and in 4 out of 4 MPG tasks. Thus, in cases when a-ops regret is not very low, it is still much better than that of the alternative methods, and when a-ops does not outperform some of the methods, its regret remains very low. As a result, despite variability in different tasks, a-ops is a very reliable method for policy selection.

MPG

For MPG, OPE performs exceedingly well on 3 of 4 tasks, getting regret close to zero for 2 tasks. Nevertheless we manage to perform about as well or better on all of the tasks. A-ops makes the most improvement in the Slide task.

DM Control Suite

In manipulator tasks, no method achieves a regret as low as in other tasks. We believe that the main reasons for this are 1) low performance of the initial OPE estimates and 2) the very skewed distribution of episodic returns of all policies where most returns are close to 0.

In general, the main factors that affect the viability in the a-ops performance are 1) the distribution of FQE estimates vs ground truth, 2) the variance of returns of a single policy, 3) variance in the returns across available policies. We will add the more detailed discussion of the results in the paper revision and we will include the illustrations that show the progression of methods in different tasks.

Some reviewers (**68yo**, **bUXf**) stated that more detailed discussion of the method's limitations would be useful. Here we discuss several of the identified limitations and future work directions to mitigate them.

Reviewers **bUXf, rn9y: Safety concern in online interaction.**

bUXf: *"if deploying on real hardware you could break the robot"*

rn9y: *"online interactions with the environment is not available for reasons including ethical or cost concerns"*

A-ops approach could be of use in many safety critical applications. This work does not deal with this question explicitly as we assume safety constraints to be implemented directly on the hardware. It means that if an unsafe policy is attempted, the safety controller would terminate policy before reaching the end of an episode, resulting in low reward and thus discouraging a-ops to try this policy again.

Alternatively, we can identify a potential solution that can be incorporated in the a-ops method by extending it with the safe exploration techniques in Bayesian optimization. First, we would need to adapt OPE to estimate the probability of violating a safety constraint using the offline dataset. Then, we would need to consider two GP models, one for modeling the real policy value and the other for the safety constraints. Applying techniques similar to [60] would allow us to find the best policy with low probability of violating the safety constraint in a data efficient manner.

Reviewers **6fix, bUXf: Use of online trajectories.**

6fix: *"The additional data collected during online interactions could be used to improve the OPE estimates since they are particularly on-policy samples for a candidate policy?"*

bUXf: *"Aside from using the episode returns, A-OPS does not leverage any of the trajectory data "*

In this work, we explore the policy structure through the offline dataset and treat every new trajectory only as an observation of noisy policy values. This allows us to apply the Bayesian optimization formulation in a simple way.

We acknowledge that more information could be used from the new trajectories. The obvious way to extend the a-ops is to use them to further improve OPE estimates, GP kernel, or policies themselves. However, including this information would make the solution much more complex as OPE estimates, GP model and the policies need to change at every step. This is no longer a subject for classic Bayesian optimization. Moreover, we cannot neglect the additional computational cost of these updates which could be considerably high even compared to the cost of new data collection.

With these challenges in mind, we consider the clean Bayesian optimization formulation as our first attempt to address this new problem setting of a-ops. Even when we use only trajectory returns, we obtain competitive results compared to multiple baselines. We hope this work will spark some interest in the community to explore novel uses of the online interactions combined with offline data in creative ways.

---

### Decision · Program_Chairs · 2021-09-27

**Decision:**

Accept (Poster)

**Comment:**

The reviewers felt this paper would be a nice contribution to the conference. It introduces a novel problem setting of active offline policy selection problem formulation, which relies both on logged data and limited online interactions with the environment.  The authors have done a good job replying to all concerns during the discussion.  To strengthen the contribution, authors are encouraged to open source components of the approach.